# High voltage electrolytes for lithium-ion batteries with micro-sized silicon anodes

Ai-Min Li[1], Zeyi Wang[1], Travis P. Pollard [2], Weiran Zhang[1], Sha Tan[3], Tianyu Li[4], Chamithri Jayawardana[5], Sz-Chian Liou[6], Jiancun Rao[6], Brett L. Lucht[5], Enyuan Hu [3], Xiao-Qing Yang [3], Oleg Borodin [2] ✉ & Chunsheng Wang [1] ✉

Micro-sized silicon anodes can significantly increase the energy density of lithium-ion batteries with low cost. However, the large silicon volume changes during cycling cause cracks for both organic-inorganic interphases and silicon particles. The liquid electrolytes further penetrate the cracked silicon particles and reform the interphases, resulting in huge electrode swelling and quick capacity decay. Here we resolve these challenges by designing a high-voltage electrolyte that forms silicon-phobic interphases with weak bonding to lithium-silicon alloys. The designed electrolyte enables micro-sized silicon anodes (5 μm, 4.1 mAh cm$^{-2}$) to achieve a Coulombic efficiency of 99.8% and capacity of 2175 mAh g$^{-1}$ for >250 cycles and enable 100 mAh LiNi$_{0.8}$Co$_{0.15}$Al$_{0.05}$O$_2$ pouch full cells to deliver a high capacity of 172 mAh g$^{-1}$ for 120 cycles with Coulombic efficiency of >99.9%. The high-voltage electrolytes that are capable of forming silicon-phobic interphases pave new ways for the commercialization of lithium-ion batteries using micro-sized silicon anodes.

Li-ion batteries (LIBs) have come to dominate the portable electronics landscape since their commercialization[1–4]. However, the expanded use of LIBs in electric vehicles and grid storage has necessitated the adoption of high energy-density materials including Ni-rich cathodes and Li metal or Si anodes[5–7]. Si has a high theoretical capacity (~3579 mAh g$^{-1}$ of Li$_{15}$Si$_4$ vs ~372 mAh g$^{-1}$ of LiC$_6$), low electrochemical potential (~0.3 V vs Li/Li$^+$), and is naturally abundant[8–11]. However, the large volume change (280%) during lithiation/de-lithiation induces pulverization of Si particles, further degrading Coulombic efficiency (CE) and resulting in poor cell cycle life[12,13]. Strategies including the use of nano-Si particles or wires[12,13], highly-elastic binders[14–18], and Si/graphite composite materials[19–22] have been reported to overcome the stability challenges of micro-sized Si (μSi) electrodes. Nano-Si anodes in carbonate electrolytes can achieve 1000 cycle life due to significantly reduced volume change of nano-Si during lithiation/delithiation cycles. However, nano-Si anodes suffer from high cost, low-

taping density, low calendar life, and pre-lithiation requirements[23], which limit the nano-Si application for sustainable LIBs. Recently, we revisited the μSi electrodes to reveal its capacity decay mechanism[24]. The pulverization of μSi is unavoidable due to the large volume change during lithiation/de-lithiation cycles. However, the reason for the capacity decay is not induced by μSi pulverization but rather the cracking of the solid electrolyte interphase (SEI), which allows electrolytes to penetrate the cracked Si particles and form new SEI, further isolating the cracked Si (loss of contact) (Fig. 1a). The reason for SEI cracking during cycling is because the organic-inorganic SEI formed in commercial carbonate electrolytes strongly bonds to Li$_x$Si and experiences the same volume change as μSi electrodes, thus the SEI cracks synchronously as μSi particles (Fig. 1a). An ideal electrolyte is expected to form silicon-phobic LiF SEI that weakly adheres to μSi particles, which allows the Li$_x$Si phase to shrink without damaging the SEI, leading to full μSi capacity utilization with no electrolytes

[1]Department of Chemical and Biomolecular Engineering, University of Maryland, College Park, MD 20740, USA. [2]Battery Science Branch, DEVCOM Army Research Laboratory, Adelphi 20783 MD, USA. [3]Chemistry Division, Brookhaven National Laboratory, Upton, NY 11973, USA. [4]Department of Chemistry and Biochemistry, University of Maryland, College Park, MD 20740, USA. [5]Department of Chemistry, University of Rhode Island, Kingston, RI 02881, USA. [6]Maryland Nanocenter, University of Maryland, College Park, MD 20740, USA. ✉e-mail: oleg.a.borodin.civ@army.mil; cswang@umd.edu

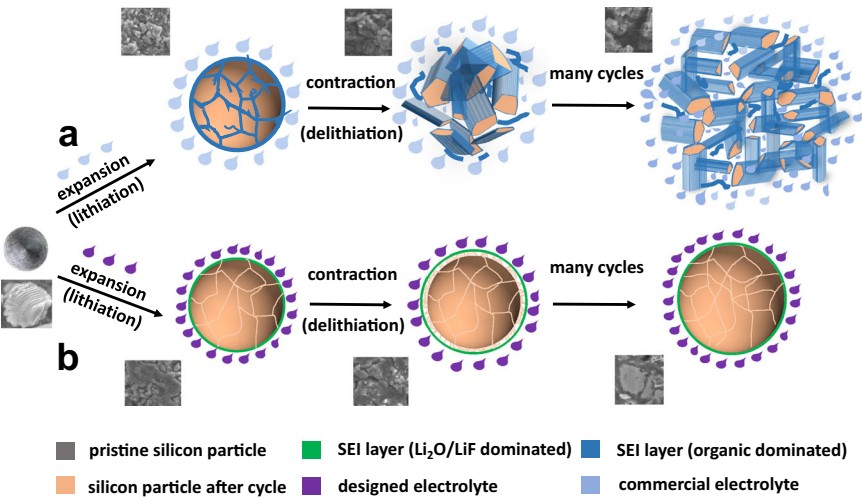

**Fig. 1 | The reversibility of SiMPs anodes covered with silicon-phobic Li$_2$O-LiF rich SEI or silicon-philic organic-rich SEI during lithiation/de-lithiation cycles. a** Schematic illustration of SiMPs electrodes cycled in conventional carbonate electrolytes that form silicon-philic organic-inorganic SEI with strong bonding to Si. **b** Schematic illustration of SiMPs electrodes cycled in the designed electrolytes that form silicon-phobic Li$_2$O-LiF SEI with weak bonding to Si.

penetrating the cracked μSi particles (Fig. 1b). The reduction of fluorinated inorganic salt generates LiF-rich inorganic SEI, while the reduction of organic solvent generally forms organic-rich organic/inorganic mixed SEI. To promote anion reduction, high-concentration electrolytes were used to increase anion numbers in the Li$^+$ solvation shell[1,25,26]. To suppress the reduction of organic solvent, weakly solvating solvents were used to reduce the solvent numbers in Li$^+$ solvation shell[27], and highly stable ether solvents such as 1,2-dimethoxyethane (DME)[28] and tetrahydrofuran (THF)[24] were used to lower the reduction potential of the solvents. However, the low oxidation potential, low boiling point, and high flammability of ether-based electrolytes reduced the cell energy/power density and operation safety. To enhance the oxidation potential and safety of the electrolytes, solvents with a high oxidation potential should be considered. However, it will raise the organic component in SEI due to the high reduction potential of solvents, reducing the CE and cycle life of the μSi electrodes. The employ of fluorinated carbonate solvents can increase the LiF content in the SEI, but it will also increase the organic components[29,30]. For example, the high-voltage all-fluorinated carbonate electrolytes (1.0 M LiPF$_6$ in FEC (fluoroethylene carbonate)-FEMC (2, 2, 2-trifluoroethyl, methyl carbonate)-TTE (1, 1, 2, 2-tetrafluoroethyl-2, 2, 3, 3-tetrafluoropropyl ether) (denoted as FFT) enable μSi anodes to achieve a CE of 99.7% when cycled at a low capacity of >1000 mAh g$^{-1}$ [29]. However, a quick capacity decay of the μSi electrodes is still observed in the all-fluorinated FFT electrolyte due to high organic content in the SEI arising from the reduction of fluorinated carbonates. To date, no electrolytes can simultaneously enable high CE on both μSi anodes and high-voltage cathodes to achieve high-energy LIBs, and no practical large pouch cell has ever been demonstrated with μSi anodes.

Herein, we report a 4.3 V sulfolane-based electrolyte consisting of 1.0 M LiPF$_6$ salt and a 2:6:2 (by volume) mixture of fluoroethylene carbonate (FEC), sulfolane (SL), and 1, 1, 2, 2-tetrafluoroethyl-2, 2, 3, 3-tetrafluoropropyl ether (TTE) for μSi‖NCA cells. The designed FEC-SL-TTE electrolytes are denoted as FST. The FST electrolytes enable the micro-sized Si (5 μm) anode with an areal capacity of 4.1 mAh cm$^{-2}$ to achieve a high capacity of 2718 mAh g$^{-1}$ with an average CE of >99.8% and full cell performance with an NCA cathode to achieve a high CE of 99.9% by forming LiF-rich cathode electrolyte interphase (CEI). The FST electrolytes also enable μSi‖NCA coin cells to achieve 148 mAh g$_{NCA}$$^{-1}$ capacity with 81% retention after 200 cycles, making the best performing μSi full cell to date. We further demonstrate large-scale 100 mAh high-capacity pouch cells with a long cycle life of 120 cycles

under realistic conditions (temperature, pressure, C rate), which is the first practical pouch cell demonstration ever reported with SiMPs (Supplementary Table 1). The success of 4.3 V FST electrolytes for μSi‖NCA cells is attributed to the sulfolane (SL) solvent, which has a high oxidation potential and forms inorganic Li$_2$O SEI with minimal organic components during reduction[31]. Similar to LiF, Li$_2$O is also silicon-phobic, enabling the micro-sized SiO anodes to achieve a long cycle life[32–34]. Inheriting the properties of LiF and Li$_2$O, the Li$_2$O-LiF composite SEI has weak binding to Li$_x$Si alloy, enabling SiMPs to reversibly expand/contract inside the SEI shell and achieve a long cycle life (Fig. 1b). The high ionic-to-electronic conductivity ratio in Li$_2$O-LiF SEI also decreases the area specific resistance (ASR), reducing the required SEI thickness needed to block electron transfer through SEI[35]. In addition, the FST electrolytes are also non-flammable, which further enhances cell operation safety.

## Results and discussions
### Electrolyte design for μSi anodes
The electrolytes for high voltage LIBs using μSi anodes should meet several stringent requirements: (1) enable the formation of a silicon-phobic inorganic SEI (such as LiF or Li$_2$O-LiF composite SEI) that has high interfacial energy and weak binding to Li$_x$Si alloy phase; (2) enable the formation of LiF-rich cathode electrolyte interphase (CEI) to support high voltage/high capacity cathodes (such as NCA or NMC); (3) have a high ionic conductivity (>10$^{-3}$ S cm$^{-1}$); and (4) be nonflammable. The designed FST electrolytes satisfy all the above harsh requirements. The key for electrolyte design here is to enhance the inorganic LiF/Li$_2$O components while minimizing the organic counterparts in the formed SEI/CEI. As stated earlier, the reduction of fluorinated inorganic salts (LiPF$_6$, LiFSI, etc) forms LiF-rich inorganic SEI, while the reduction of organic solvents will form both organic and inorganic SEI. To reduce the organic components in the SEI, the reduction of solvent should form more inorganic Si-phobic compounds (Li$_2$O, LiF, etc) and fewer organic species or can be re-dissolved in the mother electrolytes, leaving inorganic contents accumulated in the final ceramic SEI. SL is a highly polar aprotic solvent (dielectric constant of 43.4 at 303.2 K) with high thermal and anodic stability windows[36]. Density functional theory (DFT) calculations suggest that when SL is bound to two Li$^+$, it reduces at 1.3–2 V vs Li/Li$^+$ to form Li$_2$O (Supplementary Fig. 1) at the same potential range as LiF is formed with the reduction of Li$^+$(FEC) and TTE. Molecular dynamics (MD) simulation of FST electrolytes discussed below show that ~4% of SL are indeed coordinated by 2 Li$^+$ and would

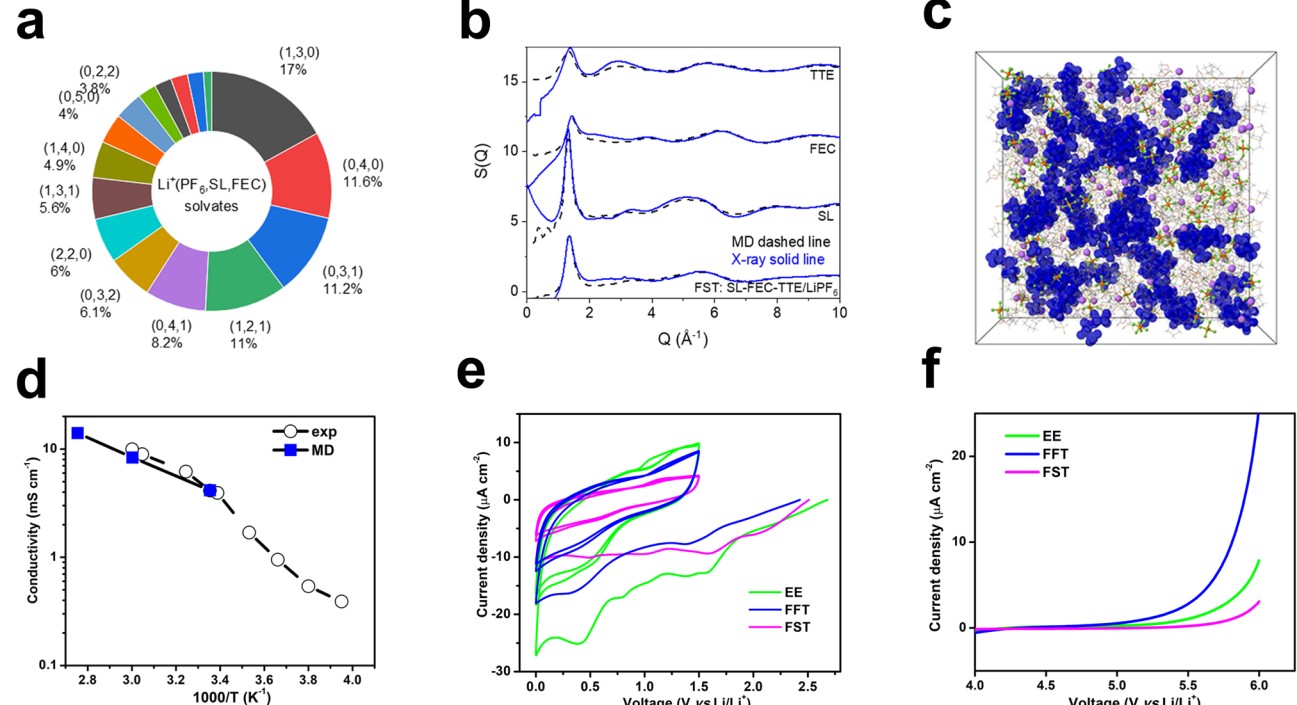

**Fig. 2 | Solvation structure, conductivity, and electrochemical stability of the investigated electrolytes. a** Distribution of the Li$^+$ solvates of FSI electrolytes from MD simulations, showing a percent for a specific solvate Li$^+$(PF$_6^-$, SL, FEC), only the solvates with populations above 1% are shown for clarity. **b** X-ray weighted structure factor for TTE, FEC, SL solvents, and FST electrolytes from both MD simulations and experiments at room temperature except for SL (30 °C). **c** A snapshot of the MD simulation cell at 25 °C with FEC, SL solvents shown as wireframes (gray for C, red for O, green for F, and yellow for S) and TTE diluent highlighted using blue iso-surface, other molecules are shown in the ball-and-stick model where purple balls represent Li$^+$ cations and green/brown balls indicate PF$_6^-$ anions. **d** Ionic conductivity of FST electrolytes from experiments and MD simulations. **e** Cathodic stability of three electrolytes measured using cyclic voltammetry in Li||Cu half cells, the first scan starts from open circuit potential to 0 V vs Li/Li$^+$, the following scans are between 1.5 to 0 V vs Li/Li$^+$. **f** Anodic stability of three electrolytes measured using linear scanning voltammetry in Li||Al half cells. The scan rate for CV and LSV tests is 0.5 mV s$^{-1}$. Source data are provided as a Source Data file.

yield Li$_2$O as a result of the SL(Li$^+$)$_2$ reduction, suggesting SL reduction forms Li$_2$O to supplement inorganic LiF-rich content in the SEI (Supplementary Fig. 1). In addition, SL has a high solubility for organic SEI and is nonflammable. Formulated with fluorinated FEC and TTE solvents, the FST electrolytes can simultaneously support for both μSi anodes and high-voltage NCA cathode with high cell operation safety.

### Solvation structure and properties of the studied electrolytes

The ion coordination environments in 1.0 M LiPF$_6$/EC-EMC (EE), 1.0 M LiPF$_6$ in FEC-FEMC-TTE (FFT), and 1.0 M LiPF$_6$ in FEC-SL-TTE (FST) electrolytes were characterized using Raman and multi-nuclear NMR ($^7$Li- and $^{19}$F-) spectroscopies. Raman spectra around 740–750 cm$^{-1}$ probe PF$_6^-$ anion environment due to the blue shift of this Raman band upon Li$^+$ complexation[37]. The magnitude of the shift, however, depends on the details of Li$^+$ binding to PF$_6^-$ anion (monodentate vs bidentate), complicating the interpretation of the spectra[38]. Raman spectra for FFT indicate stronger aggregation than EE electrolytes (Supplementary Fig. 2). Interpretation of FST spectra is complicated because the peaks around 750 cm$^{-1}$ could correspond to either anion coordinated to one or multiple Li$^+$ and to SL/Li$^+$ (Supplementary Fig. 2a). Therefore, in-situ NMR was used to distinguish PF$_6^-$/Li$^+$ pairing from SL/Li$^+$. The upfield shift observed in the $^7$Li-NMR spectra from EE to FFT to FST is consistent with increasing ion-pairing (EE to FFT) and replacement of stronger Li-SL contacts (FST) compared to Li-(PF$_6^-$) (FFT) (Supplementary Fig. 2b). Likewise, an upfield shift in $^{19}$F spectra is observed from EE to FFT, though it is shifted downfield in FST electrolytes (Supplementary Fig. 2c), suggesting that PF$_6^-$/Li$^+$ coordination increases in all-fluorinated FFT electrolytes but decreases when FEMC is replaced by SL due to stronger Li$^+$/SL binding energy as shown in Supplementary Fig. 3. Consequently, SL has the highest solvation ability and likely dominates the Li$^+$ solvation shell.

MD simulations were used in conjunction with pair distribution functions obtained from the synchrotron X-ray source to further characterize the solvation structure of the FST electrolytes (Fig. 2a–c). In accord with the Li$^+$(SL) > Li$^+$(FEC) > Li$^+$(FEMC) > Li$^+$(TTE) binding energy trends from DFT shown in Supplementary Fig. 3, MD simulations predict the Li environment being SL-rich and Li$^+$(SL)$_4$, Li$^+$(SL)$_3$(FEC), LiPF$_6$(SL)$_3$ and LiPF$_6$(SL)$_2$(FEC) being the most probable Li$^+$ solvates in FST electrolytes (Fig. 2a). The Li$^+$ cation is primarily coordinated by 2.9 SL, 0.8 FEC and 0.7 PF$_6^-$ anions on average with a negligible presence of TTE (Supplementary Table 2) corresponding to the radial distribution functions (RDFs) shown in Supplementary Fig. 4a. Importantly for enabling the LiF formation as a result of (Li$^+$)$_n$(PF$_6^-$) and Li$^+$(FEC) reduction, non-negligible Li-F(FEC) and Li-F(PF$_6$) contacts are observed as shown in Supplementary Fig. 4b.

The predicted X-ray weighted structure factor from MD simulations for TTE, FEC, SL solvents, and FST electrolytes agreed well with the measured ones further validating our electrolyte structure predictions (Fig. 2b). The representative solvates and aggregation of the TTE diluent in the simulation box are shown in Fig. 2c and Supplementary Fig. 5, indicating the existence of Li$^+$ ion conducting SL-rich and TTE-rich domains.

The physical and electrochemical properties of the solvents and three electrolytes are listed in Table 1. The ionic conductivity of FST electrolytes at different temperatures was measured and the conductivity above room temperature agreed well with the MD simulation predictions (Fig. 2d, Table 1, Supplementary Table 2). The FST electrolytes have a high ionic conductivity of >4 mS cm$^{-1}$ at 25 °C and high Li$^+$ transference numbers: 0.67 (experiment) and 0.59 (MD simulation).

**Table 1 | Properties of solvents and electrolytes at 25 °C from experiments and MD simulations**

| Compounds | Diffusion coefficients ($10^{-10}$ $m^2$ $s^{-1}$) | Boiling point (1 atm) | Viscosity (cP) | Density (g mL$^{-1}$) |
|---|---|---|---|---|
| EC | – | 248.0 | Solid | Solid |
| EMC | – | 110 | 0.65 | 1.01 |
| FEC | 1.27 | 210 | 3.85 | 1.41 |
| FEMC | – | 92 | 1.42 | 1.31 |
| TTE | 1.01 | 93.2 | 1.23 | 1.53 |
| SL | 0.88 | 285 | solid | solid |
| **Electrolytes** | **ionic conductivity (mS cm$^{-1}$)** | **Li$^+$ transference number** | **Viscosity (cP)** | **Density (g mL$^{-1}$)** |
| EE | 10.17 | 0.37 | 3.12 | 1.27 |
| FFT | 5.12 | 0.50 | 2.15 | 1.51 |
| FST tested (MD) | 3.93 (3.6-4.2)$^a$ | 0.67 (0.59) | 16.15 (16.8) | 1.47 (1.42) |

Note: EE: 1 M LiPF$_6$-EC-EMC (1:1 by volume); FFT: 1 M LiPF$_6$-FEC-FEMC-TTE (2:6:2 by volume); FST: 1 M LiPF$_6$-FEC-SL-TTE (2:6:2 by volume).
$^a$MD simulations predict conductivity of 3.6 mS cm$^{-1}$ before finite simulation cell correction and 4.2 mS cm$^{-1}$ after the finite simulation cell correction was applied.

The cathodic and anodic electrochemical stabilities of the three electrolytes were also determined by cyclic voltammetry (CV) and linear sweep voltammetry (LSV), respectively. In the cathodic scans, compared to FFT, the FST electrolytes effectively passivated the Cu electrode after the initial scan, which largely reduced the current density in the following scans due to the formation of Li$_2$O-LiF SEI (Fig. 2e). Since SL is also an effective electrolyte component for high-voltage cathode batteries[31], the introduction of SL into the FST electrolytes also boosts its oxidation stability with no obvious current increase observed up to 5.5 V in the Li||Al half cells, a value even higher than that of the all fluorinated FFT electrolytes (Fig. 2f). Moreover, compared to the highly flammable EE electrolytes (Supplementary Movie 1, Supplementary Fig. 6a), the fluorinated FFT electrolytes showed suppressed flammability (Supplementary Movie 2, Supplementary Fig. 6b), and the designed FST electrolytes demonstrated the best flame retardant performance among the three electrolytes due to the use of SL solvent (Supplementary Movie 3, Supplementary Fig. 6c)[39], which offers improved battery operation safety (Supplementary Note 1)[29,40].

## SEI composition on SiMPs

Similar to 1.0 M LiPF$_6$ in ethylene carbonate−dimethyl carbonate (EC-DMC)[24,41], 1.0 M LiPF$_6$/EC-EMC (EE) electrolytes also contain ~60% of solvent-separated ion pairs (SSIPs), only 40% of contacted ion pairs (CIPs), and few ionic aggregates (AGGs). The reduction of CIPs in the traditional carbonate solvents occurs at potentials close to that of the pure EC and EMC/DMC solvents, forming a mixed organic and inorganic SEI with large separate domains[41]. The fluorination of the carbonate solvents has been attested to enrich LiF content in the SEI components, both on lithium metal surface[40] and silicon electrodes[29]. However, the reduction of fluorinated carbonate solvents also inevitably leads to organic components in the SEI as well, limiting the cycling CE of µSi anodes in the fluorinated electrolytes[29]. DFT calculations demonstrated that FEC in 1.0 M LiPF$_6$ in FEC-SL-TTE (FST) has the highest reduction potential (-1.9 V) when its fluorine is close to Li$^+$ (Supplementary Figs. 1, 7), leading to LiF formation and initial FEC polymerization. The main Li$^+$(FEC) reduction when Li$^+$ is away from fluorine occurs at much lower potentials (-1 V vs Li/Li$^+$). Without Li$^+$ coordination, the reduction of TTE occurs in the range of 1−1.6 V (see Supplementary Fig. 8). Li$^+$(SL) reduction occurs closer to 0−0.3 V with minimal deformation of the SL; however, recent work by Zheng et al.

suggested that the reduced SL• radical has a much smaller barrier of ring opening than for cycling carbonates such as PC[39]. If this ring opening occurs simultaneously with SL reduction, the reduction potential will increase to -1.6 V (Supplementary Fig. 1) and may serve as the precursor for Li$_2$SO$_x$ species in the SEI. Alternatively, ~4% of SL molecules are coordinated by 2 Li$^+$, which allows direct Li$_2$O formation at potentials near 2 V. The reduction of [Li$_2$SL•]$^+$ ring-opened radical, however, does not release Li$_2$O as loss of oxygen from the terminal SO$_2$ group is not stable. Similar reduction potentials especially for FEC and SL indicates that LiF and Li$_2$O will form simultaneously, resulting in the formation of the Li$_2$O-LiF SEI. SL additionally assists in dissolving organic/polymeric species resulting from the reduction of the solvents. Because LiEMC is a typical organic component in SEI[42], the solubility of LiEMC SEI in EE, FFT, and FST electrolytes was evaluated through $^1$H-NMR spectra (Supplementary Figs. 9, 10, Supplementary Note 2). Neither EE nor FFT electrolytes dissolve LiEMC while it can be dissolved in the FST electrolytes, leaving the Li$_2$O-LiF dominated SEI, which is further confirmed by XPS spectra below.

The SEI composition on the SiMP electrodes after cycling in different electrolytes was characterized using XPS with an Ar$^+$ sputtering time (0 s, 60 s, 120 s, 180 s, 300 s, and 600 s). The SiMP electrodes were washed with corresponding mother solvents (without salt) before the XPS analysis. Sample preparation and transferring were performed under an inert Ar atmosphere to avoid any contamination from the air. Figure 3 shows the SEI composition on the SiMP electrodes after 50 plating/stripping cycles at 1 mA cm$^{-2}$ and 4.1 mAh cm$^{-2}$ in FST, FFT, and EE electrolytes (full spectra are shown in Supplementary Figs. 11–13). The outer and inner layer of SEI formed in EE and FFT electrolytes mainly consist of organic species (C-O/C=O peak, ~286.5 eV, C-H/C-C peak, ~284.8 eV) (Fig. 3a, Supplementary Fig. 14a, b). In comparison, the FST-SEI has a thinner C-H/C-C peak with a much weaker C-O/C=O intensity than that in EE/FFT electrolytes. Organic species were primarily found in the outer FST-SEI layer and disappeared after 300 s sputtering while the inner layer of FST-SEI was almost exclusively Li$_2$O-LiF (Fig. 3b, Supplementary Fig. 14c). In the O1s spectra, the FST-SEI showed a much higher Li$_2$O intensity compared to that in FFT-SEI, and only a negligible Li$_2$O signal was noticed in the EE-SEI (Fig. 3b). Instead, the Li$_2$CO$_3$ and LiOR signals increased largely for both FFT and EE electrolytes. This result validates that FST electrolytes could promote the formation of Li$_2$O in the SEI by sulfolane reduction as suggested by the MD simulation (Supplementary Fig. 1). A similar decrease trend was found for the LiF signal in the F1s spectra from FST to FFT and EE electrolytes (Supplementary Fig. 11). The simultaneous formation of Li$_2$O and LiF in FST electrolytes leads to the desired Li$_2$O-LiF composite SEI that will be beneficial for the long cycle of SiMPs. The Li$_2$CO$_3$ region also widens in FST-SEI, suggesting the presence of Li$_2$SO$_x$ species as confirmed in the S2p spectra (Fig. 3b, Supplementary Fig. 13). The F-content is abundant throughout the etching process for FST-SEI, confirming that a highly inorganic-rich Li$_2$O-LiF SEI layer is present. The presence of crystalline LiF and Li$_2$O in SEI was also verified by the Fast Fourier Transform (FFT) patterns obtained from high-resolution transmission electron microscopy (HRTEM) imaging (Supplementary Fig. 15). The relatively high ratio of F content in FFT-SEI is also in good agreement with the SEI formed on the Li metal anode[40]. A comprehensive discussion on the SEI structure formed in different electrolytes can be found in Supplementary Note 3.

## Beneficial of Li$_2$O-Li composite SEI towards SiMPs

Adhesion of SEI components to Li$_x$Si alloy phase plays a critical role in stabilizing the SiMP anodes. The adhesion of SEI components can be reflected by the Work of Separation (WoS). The WoS for Li$_2$O LiF and Li$_2$CO$_3$ to Li$_x$Si was calculated using molecular modeling, where Li$_2$CO$_3$ is used as a reference. A low WoS value corresponds to a high interface energy ($E_{int}$) (Supplementary Note 4). Figure 4a shows that LiF and Li$_2$O have lower WoS values (<0.33 J m$^{-2}$) to Li$_x$Si (Li$_{15}$Si$_4$, Li$_{12}$Si$_7$ and LiSi)

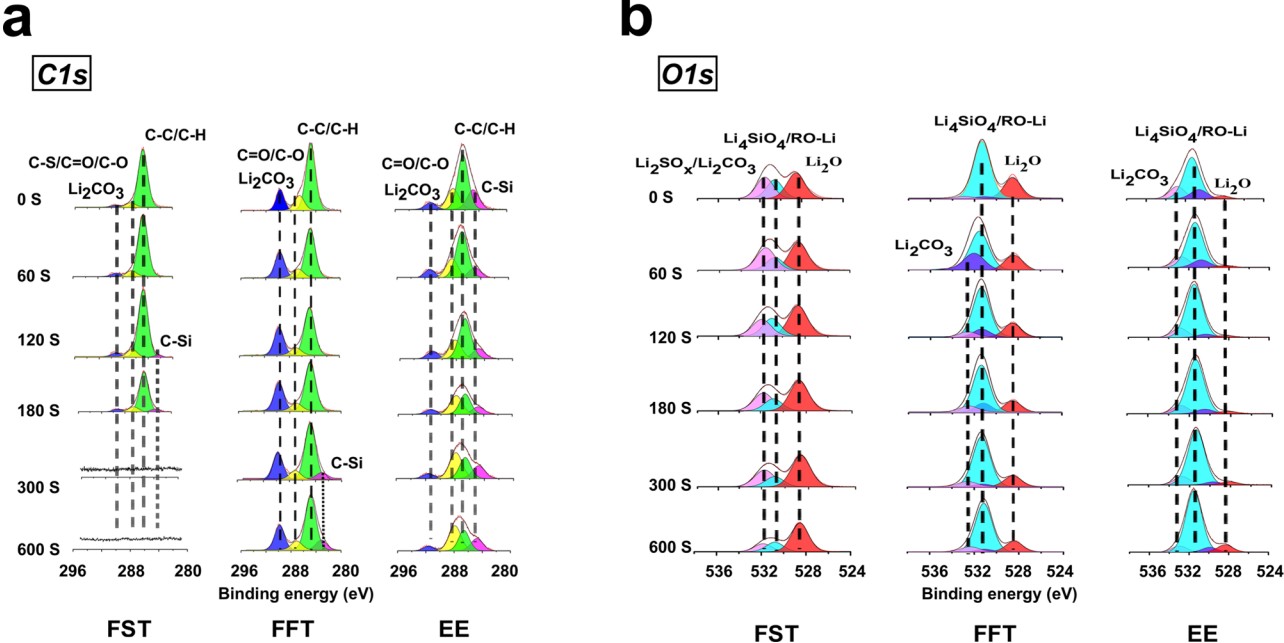

**Fig. 3 | SEI chemical composition by XPS measurement on μSi electrodes after 50 cycles in Li∥μSi half cells with different electrolytes.** *C1s* (**a**) and *O1s* (**b**) spectra are displayed in columns, which show the corresponding depth profiling results (from left to right, being FST, FFT, and EE, respectively). The relative intensity for all spectra was shown in arbitrary units (a.u.) without labeling the *y*-axis for clarity. All the XPS results were fitted with CasaXPS software. The binding energy was calibrated with *C1s* at 284.8 eV. Only the C, O, F, and Si atomic ratios are shown in the stacked columns for clear comparison, full data can be found in Supporting Information (Supplementary Figs. 11–14).

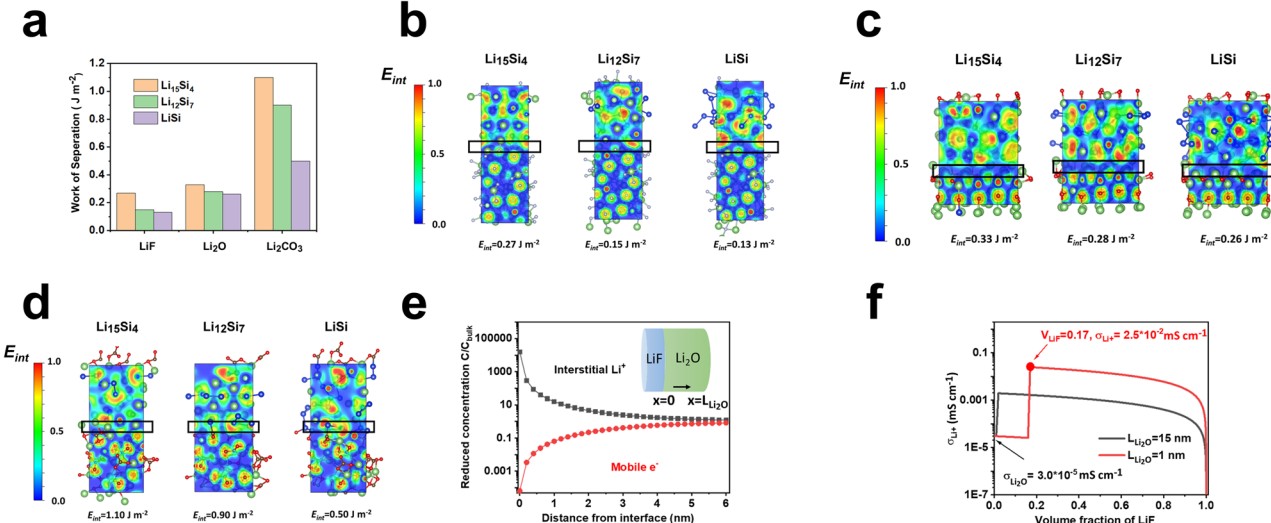

**Fig. 4 | Effect of LiF/Li₂O/Li₂CO₃ SEI on the LiₓSi alloy anodes. a** Work of separation for LiF|LiₓSi, Li₂O|LiₓSi, and Li₂CO₃|LiₓSi interfaces. **b**–**d** Electron localized function and $E_{int}$ between the LiₓSi (Li₁₅Si₄, Li₁₂Si₇, and LiSi) alloys and major SEI components (LiF, Li₂O and Li₂CO₃) with the scale bar (from 0.0 to 1.0) shown on the left. **b** LiF. **c** Li₂O. **d** Li₂CO₃. **e** The normalized concentration profile of interstitial Li⁺ and mobile electron Li₂O within Li₂O/LiF space charge region. The inserted scheme shows the configuration of the Li₂O/LiF space charge model. **f** The total ionic conductivity of the Li₂O/LiF composite SEI as a function of the volume fraction of LiF when the grain size of Li₂O is equal to 15 nm and 1 nm, respectively.

than Li₂CO₃ (up to 1.10 J m⁻²), indicating higher interfacial energies of LiF and Li₂O to the active silicon particles during lithiation process. Electron localization function (ELF) images of the three SEI components (LiF, Li₂O, and Li₂CO₃) to the lithiated silicon phases are shown in Fig. 4b–d. A region with an ELF value of <0.2 was observed for LiF|LiₓSi and Li₂O|LiₓSi interfaces, referring to the low chemical bondings at the interface. In contrast, the ELF value at the Li₂CO₃|LiₓSi interface varies from 0 to 0.9, corresponding to the formation of mixed ionic and covalent bonds. The Li₂O and LiF have high $E_{int}$ to LiₓSi, and the Si-phobic Li₂O-LiF SEI suffer less stress during the large volume change of SiMPs.

In addition to SEI stabilization, the synergetic effects of LiF and Li₂O also increase the Li-ion conductivity and reduce electron leakage by promoting space charge accumulation along their interfaces (Fig. 4e, f, Supplementary Fig. 16). The interstitial defect formed within the lattice Li⁺ ion between LiF and Li₂O was found to boost the

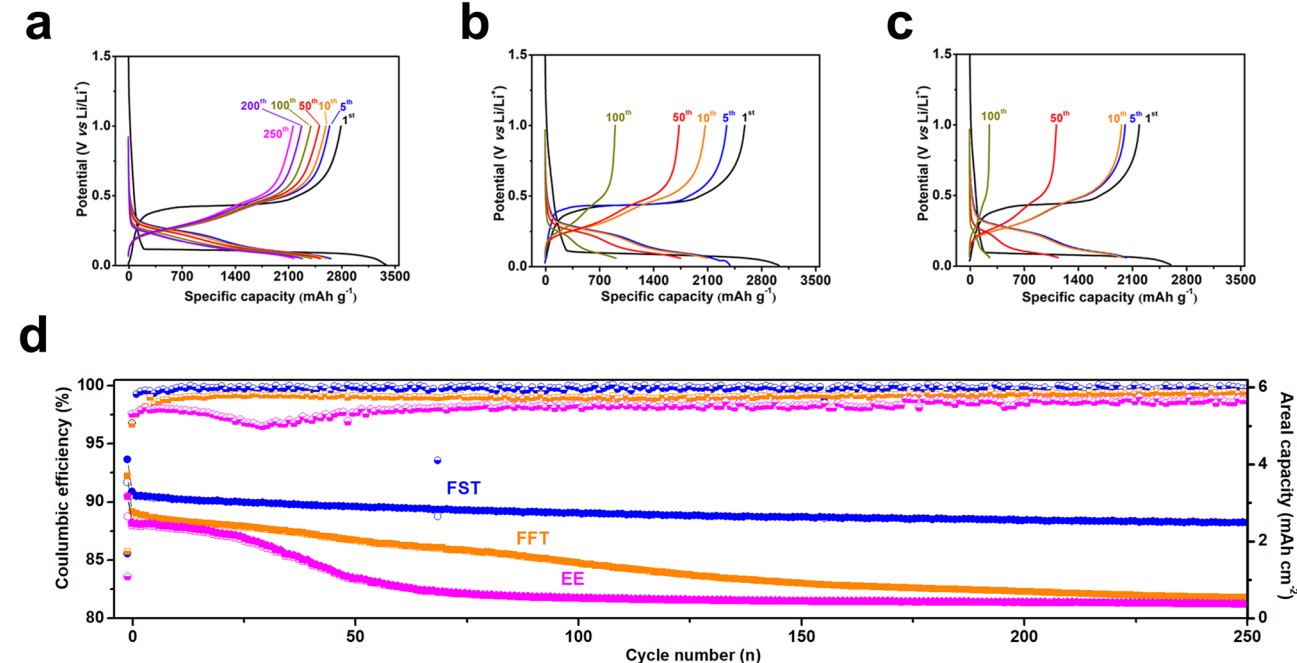

**Fig. 5 | Cycle performance of SiMP electrodes in Li∥μSi half cells. a–c** Typical charge/discharge profiles of the SiMP electrodes cycled in different electrolytes. **a** FST. **b** FFT. **c** EE. **d** Cycling stability and CEs of SiMPs cycled in FST and reference electrolytes; the cycle rate is C/4 with the first formation cycle at C/20. Source data are provided as a Source Data file.

interstitial $Li^+$ defect concentration in $Li_2O$ lattice near the $LiF$-$Li_2O$ interface up to $10^4$ times and reduce the electron concentration by a factor of $10^{-4}$ compared to that of the bulk $Li_2O$ (Fig. 4e). According to a simplified space charge model (details see Supplementary Note 5), when only 5% by volume of LiF was added to $Li_2O$ with a grain size of 15 nm, the ionic conductivity of the SEI increased from $3.0 \times 10^{-5}$ mS $cm^{-1}$ of $Li_2O$ to $2.0 \times 10^{-3}$ mS $cm^{-1}$ in the $Li_2O$-LiF composite (Fig. 4f). Further reducing the grain size of $Li_2O$ and increasing the amount of LiF can generate more $Li_2O$-LiF interface and improve the contribution of space charge effects to total conductivity. Based on this, the total ionic conductivity of $Li_2O$ and LiF composite SEI formed in the FST electrolytes was predicted to be ~$2.5 \times 10^{-2}$ mS $cm^{-1}$. The interfacial calculation indicates that the high-modulus $Li_2O$-LiF film not only ensures low bonding between SEI and $Li_xSi$ phases ($Li_xSi$-phobic) but also promotes space charge accumulation along their interfaces. These effects suppress crackings of SiMPs during cycling and generate a high ionic-to-electronic conductivity ratio, reducing electron leakage and overall SEI thickness to enable high CE and long-cycle stability of SiMPs.

## Electrochemical performance of SiMPs anodes

The electrochemical performance of the 5 μm silicon electrodes with a ~1.2 mg $cm^{-2}$ mass loading was investigated in FST electrolytes between 0.05 V and 1.0 V at a current of 0.25 C in the Li∥μSi half cells. Before the performance evaluation, the μSi electrodes experienced one formation cycle between 0.005 and 1.0 V at a low current of 0.05 C. The performance of the 5 μm Si electrodes in EE and FFT electrolytes was also tested for comparison. The μSi electrodes show a high initial capacity of 4.1 mAh $cm^{-2}$ and ~3380 mAh $g^{-1}$ with initial Coulombic efficiency (iCE) of 85.6% in the formation cycle at a current density of 0.05 C, discharge cut-off potential of 0.005 V in the FST electrolytes (Fig. 5a). After the first cycle, the μSi electrodes were charged/discharged at a high current density of 0.25 C and high discharge cut-off potential of 0.05 V. The CE of the μSi electrodes increases to 96.8% at the 2nd cycle and then to 99.3% in the 3rd cycle with an average Coulombic efficiency (aCE) of 99.8% from the 2nd to

250th cycle. The 5 μm Si in FST electrolytes was able to deliver a high capacity of ~2718 mAh $g^{-1}$ at 0.25 C with a capacity retention of over 80% after 250 cycles (Fig. 5a, d). The high and stable capacity of μSi electrodes in FST electrolytes is attributed to the silicon-phobic $Li_2O$-LiF SEI. The weak bonding between $Li_2O$-LiF SEI and $Li_xSi$ core enables the SEI shell to maintain high stability during large volume changes of the inner Si core, preventing the liquid electrolytes from penetrating cracked Si particles, thus ensuring electrical connection between cracked Si particles. The simple electrolyte engineering of FST enables the SiMPs to achieve performance better than the complicated graphene confinement[19] and elastic binder[18], and comparable to the performance in low-voltage THF electrolytes[24] (Supplementary Table 1).

In sharp contrast, the SiMPs in conventional carbonate EE electrolytes can only release ~2600 mAh $g^{-1}$ capacity in the formation cycle at a rate of 0.05 C. The cell capacity quickly decreased to ~37% of its initial value in only 50 cycles (Fig. 5c) and further dropped to ~15% (250 mAh $g^{-1}$) after 100 cycles. The fast capacity decay of SiMPs in commercial carbonate EE electrolytes is attributed to the high organic component in SEI, which cannot accommodate the large volume changes of SiMPs. The CE of SiMPs was only 96–97% in the first several cycles and hovered around 98.0% after the 100th cycle (Fig. 5d). The all-fluorinated FFT electrolytes enable SiMPs to achieve an initial capacity of ~3033 mAh $g^{-1}$ with iCE of 85.7% in the formation cycle at 0.05 C but it decreases to 2390 mAh $g^{-1}$ at 0.25 C in the second cycle (Fig. 5b). The CE of 5 μm Si in FFT electrolytes increases to 99.1% in the 20th cycle with an average CE of 99.0% from the 2nd to 100th cycle, which is lower than that (99.8%) of FST electrolytes but is higher than that (97.5%) in commercial carbonate EE electrolytes (Fig. 5c). The improved CE of Li∥μSi cells in FFT electrolytes is attributed to the increase of LiF in the SEI composition. However, the organic parts from the reduction of fluorinated carbonates still hinder the robustness of the formed SEI. The low CEs of SiMPs in FFT result in continuous capacity fading to 40% in 100 cycles. In addition, the SEI resistance in the EE and FFT electrolytes shows a slight decrease from the first to the fifth cycle due to SiMP fractures with an increase in surface area[43] (Supplementary

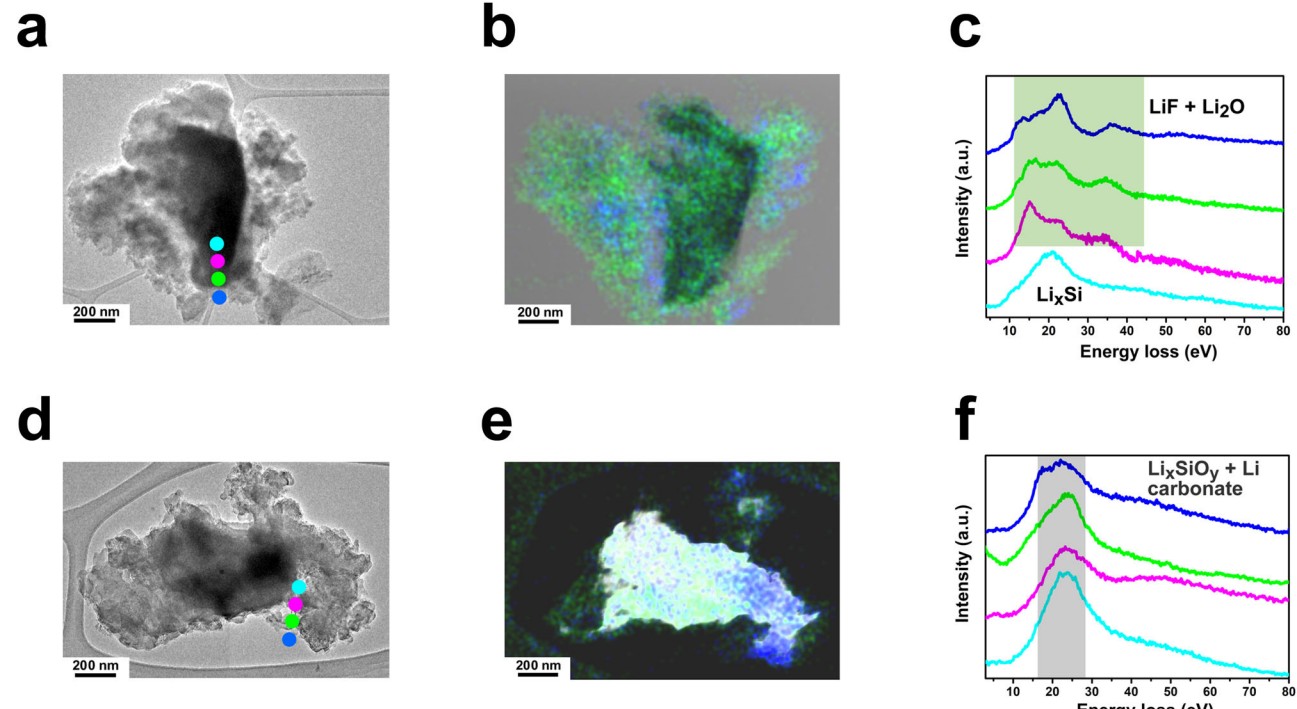

**Fig. 6 | SEI (Li₂O, LiF and Li carbonate) distribution of µSi anodes cycled with FST and FFT electrolytes. a, d** HR-TEM images, the colored dots represent the area of corresponding EELS spectral images in **c** and **f. b, e** The mapping of the selected area showing the Li₂O (green) and LiF (blue) distribution, the color contrast was adjusted for a clear comparison. **c, f** Typical EEL spectra near the surface of the µSi particles with the marked four areas in **a, d** (from surface to inner layer), a.u. (arbitrary unit) indicates the relative signal intensity. Source data are provided as a Source Data file.

Fig. 17a, b), followed by an impedance increase due to the continuous growth and thickening of the SEI on the electrodes consistent with previous reports[24]. In contrast, the thin and stable SEI formed in FST electrolytes showed small and almost-constant SEI resistance during cycling (Supplementary Fig. 17c). Since Li₂O has high interface energy against Li$_x$Si phase, replacing µSi by µSiO can further enhance the cycling stability in FST electrolytes (Supplementary Fig. 18), and even in FFT and EE electrolytes (Supplementary Figs. 19, 20). µSiO anodes not only reduce the volume change during lithiation/de-lithiation but also reduce the stress (Supplementary Note 6).

### SiMPs electrode morphology and thickness evolution

The conformal coating of Si particles by Li₂O-LiF SEI was also examined by electron energy loss spectroscopy (EELS) spectrum imaging. The signature differences in valence plasmon energy and spectral features among Li compounds in the SEI make the plasmon signals useful to distinguish them from each other easily without suffering from electron beam damage[43,44]. The EELS spectra at different locations from the surface to the interior of the SiMPs cycled in FFT and FST electrolytes were analyzed (Fig. 6). The sharp valence plasmon peaks around 13 eV, 18.4 eV with a smooth shoulder at ~34.5 eV identified the existence of Li₂O signal in SEI, while the predominant peak at 25.7 eV accompanied by a small bump of 15.3 eV is the fingerprint of LiF in the SEI layer[43,44]. For SiMPs cycled in FST electrolytes (Fig. 6a, c), the Li₂O-LiF was a homogeneous distribution on the Si particle surface with signature signals at 15 eV, 25 eV, and 35 eV, which are in good agreement with the Li₂O-LiF SEI formation mechanism supported by the molecular modeling and XPS analysis.

For SiMPs cycled in FFT electrolytes, a mixed organic–inorganic SEI with a broad peak centered around 23 eV is found for almost all the near-surface spectra, which indicates that there is neither substantial amount of Li₂O nor LiF on the surfaces (Fig. 6f). The EELS data agrees well with elemental mapping in corresponding cycled SiMPs (Fig. 6b, e,

Supplementary Figs. 21, 22). The formation of a fixed Li₂O-LiF SEI shell makes the expansion/contraction of the Li$_x$Si core more reversible and the electrode thickness remains constant after the first few charge/ discharge cycles. To validate this stability mechanism, the SiMP morphology and electrode thickness after long cycles were evaluated using scanning electron microscopy (SEM) (Fig. 7).

As shown in Fig. 7a–c, the SiMPs after charge/discharge in FST electrolytes for 200 cycles showed crack-less morphology (Fig. 7a), similar to the crack-free pristine Si with expended size and deformed shape (Supplementary Fig. 23b). Only minor fractures were found in the SiMPs electrodes. In addition, a homogeneous distribution of C, O, and F was identified in the elemental energy-dispersive X-ray spectroscopy (EDX) mapping (Supplementary Fig. 27), validating a uniform Li₂O-LiF SEI layer formation. In sharp contrast, large fractures with huge porous structures have developed in SiMPs cycled with the reference electrolytes (Fig. 7b for FFT, 7c for EE). The C, O, and F elements were found unevenly spreading over the electrodes with a high intensity of C and O for the SiMPs cycled with FFT and EE electrolytes (Supplementary Figs. 28–29), correlating to the organic-rich SEI that leads to continuous electrolyte penetration and further SiMP pulverization. The large pores in the swelled µSi electrodes lead to the loss of contact between active SiMPs and carbon black, resulting in fast capacity decay. The thickness of µSi electrodes after cycling in three electrolytes at different cycles was also measured (Fig. 7d, e). In their pristine state, the cross sections of the SiMP electrodes showed a dense packing of the silicon particles with a thickness of 18 µm (Fig. 7d, e, Supplementary Fig. 23a). After cycling, the Si electrodes cycled in FFT and EE electrolytes became loosely packed structures and the thickness continuously increased with cycling to reach 72 ± 1 µm and 113 ± 3 µm at 200 cycles, respectively (Fig. 7d, e, Supplementary Figs. 24, 25) due to the continuous formation of SEI in cracked Si. In sharp contrast, the electrodes cycled in FST electrolytes showed a more confined dense layer with a thickness of 47 ± 2 µm after 200

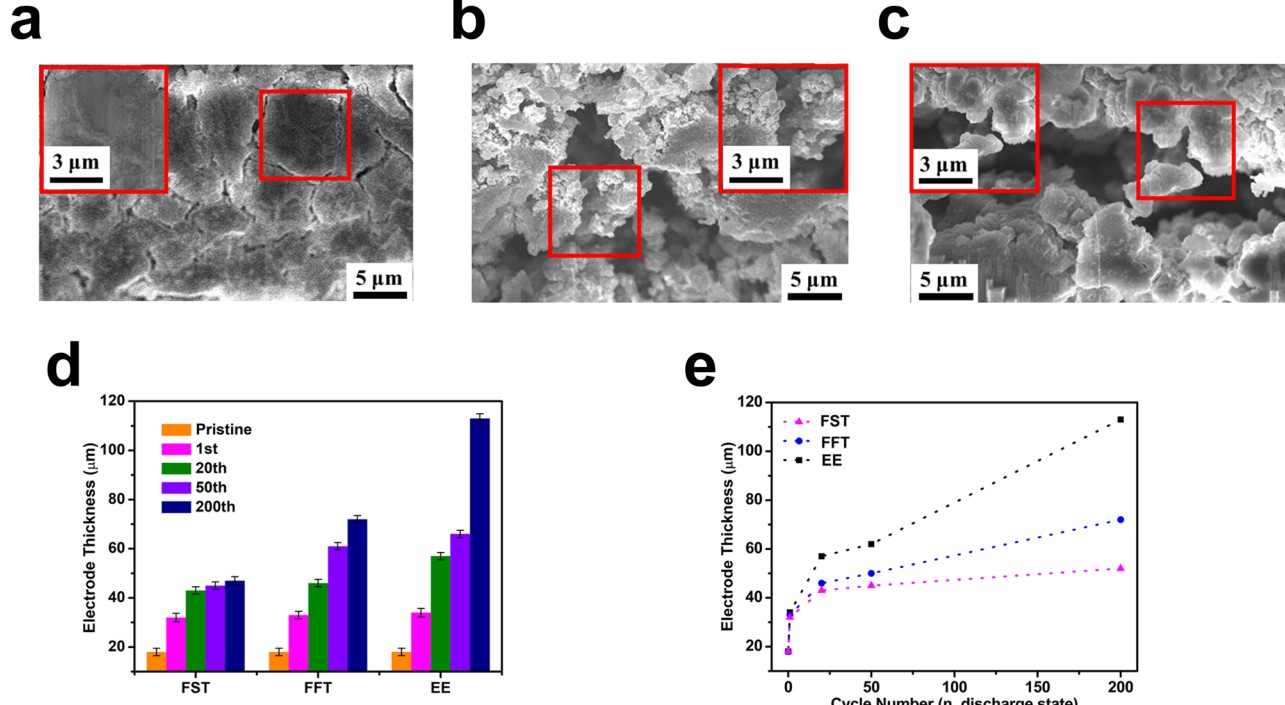

**Fig. 7 | Morphology of Si particles and electrode thickness after cycling.** Focused ion beam (FIB) cross-section SEM images of the SiMP electrodes after 200 cycles of operation in different electrolytes. The insets show the red-square highlighted area with enlarged resolution. **a** FST. **b** FFT. **c** EE. The electrode thickness evolution during the cycling with various electrolytes. **d** The histogram of thickness evolution in the three electrolytes, the error bar is defined as the average reading error from the electrode thickness measurement. **e** The SiMPs expansion trend. The dashed line here is only for the guidance of the eye. The Li‖μSi cells are cycled to a specific cycle and then stopped at the charged state to make these ex-situ measurements, detailed cross-section images can be found in Supplementary Figs. 23–26. Source data are provided as a Source Data file.

cycles, confirming the Si-phobic $Li_2O$-LiF SEI effectively prevents the electrolytes from penetrating Si particles during lithiation/de-lithiation process (Fig. 7d, e, Supplementary Fig. 26).

## μSi‖NCA full cell performance

The merits of the FST electrolytes discussed above improve the compatibility of the electrolytes with high-voltage cathodes such as NCA. Thus, we compared the performance of ~4.1 mAh cm$^{-2}$ μSi‖NCA (N/P = 1.1) full cells with EE, FFT, and FST electrolytes (Fig. 8a, b). Without any precycling nor pre-lithiation, the μSi‖NCA full cell in FST electrolytes showed an initial discharge capacity of ~183 mAh g$_{NCA}^{-1}$ with $i$CE of 80.1%. No obvious increases in the overpotentials were observed with charge/discharge cycles, which indicates that both the electrodes and their electrode/electrolyte interfaces remain stable during cycling (Fig. 8b). In contrast, under the same cell configuration and cycle conditions, only 151 mAh g$^{-1}$ and 53 mAh g$^{-1}$ initial discharge capacity are obtained for μSi‖NCA full cells cycled in FFT and EE electrolytes, respectively (Supplementary Figs. 30, 31). FST electrolytes also enable the μSi‖NCA fulls cell to achieve stable cycling (200 cycles, 81% capacity retention) with a high CE of 99.9% (Fig. 8a, blue). However. the μSi‖NCA full cells in FFT and EE electrolytes have low iCE of ~71.3% and 28.1%, respectively. The capacity of μSi‖NCA full cells in FFT and EE electrolytes also quickly decayed to <110 mAh g$^{-1}$ in 50 cycles (FFT) and <30 mAh g$^{-1}$ in 3 cycles (EE) (Fig. 8a, orange, magenta). The severe capacity decay and low CE of μSi‖NCA full cells in FFT and EE electrolytes is attributed to the continuous formation of organic SEI in cracked Si, which also increases charge/discharge voltage hysteresis (Supplementary Figs. 30, 31). Moreover, the μSi‖NCA full cells in FST electrolytes has a good rate performance due to the high ionic-to-electronic conductivity ratio of the $Li_2O$-LiF SEI (Supplementary Fig. 32). A single layer (5 cm by 5 cm) μSi‖NCA pouch cell with an areal

capacity of 4 mAh cm$^{-2}$ and N/P ratio of 1.1 was further evaluated in FST electrolytes without any pre-cycling of the anode or cathode. The practical 100 mAh μSi‖NCA pouch cell exhibited stable cycling with a high $i$CE of 81.3% and an excellent cycle CE (which approaches 99.9% after the fifth cycle) at a current density of C/5, cell pressure of 0.1 MPa, the temperature of ~25 °C (Fig. 8c, d). The large μSi‖NCA pouch full cell retained 89% of its capacity after 120 cycles in the FST electrolytes, demonstrating its superior cycle stability. This is the first-time demonstration of a μSi‖NCA pouch full with 100% depth of discharge (DoD), and the performance is the highest among the state-of-the-art μSi anode cells (Supplementary Table 1). In addition, even though small, the 0.1 MPa external pressure has been proven to be essential for the successful cycle of the μSi‖NCA pouch cells, which could ensure good electrolyte/electrode contact during the cell cycling (Supplementary Fig. 33)[45].

## CEI characterization on NCA cathodes

The CEI structure and composition on NCA cathodes were characterized with scanning transmission electron microscopy (STEM) and XPS after the 50$^{th}$ cycle at the fully discharged state in FFT and FST electrolytes. A CEI protecting layer on the primary NCA particles was observed with a CEI thickness ranging from 2–3 nm (FST, Fig. 9b) to 3–8 nm (FFT, Fig. 9a). The CEI composition on cycled NCA was further examined via X-ray photoelectron spectroscopy (XPS) (Fig. 9e, f, Supplementary Figs. 34, 35). Both CEI films formed in FFT and FST electrolytes showed high F content as evidenced by the F/C and F/O ratios of 0.36/1.3 and 0.47/1.3, respectively, indicating LiF-dominated CEI. The wide band gap (13.6 eV) and high oxidative stability (6.4 V vs Li/Li$^+$) of LiF ensured effective suppression of the parasitic reactions between the cathode surface and electrolytes[46]. The reduced M-O species (~529.5 eV, *O1s*, Fig. 9c–f) and high LiF in CEI formed in FST

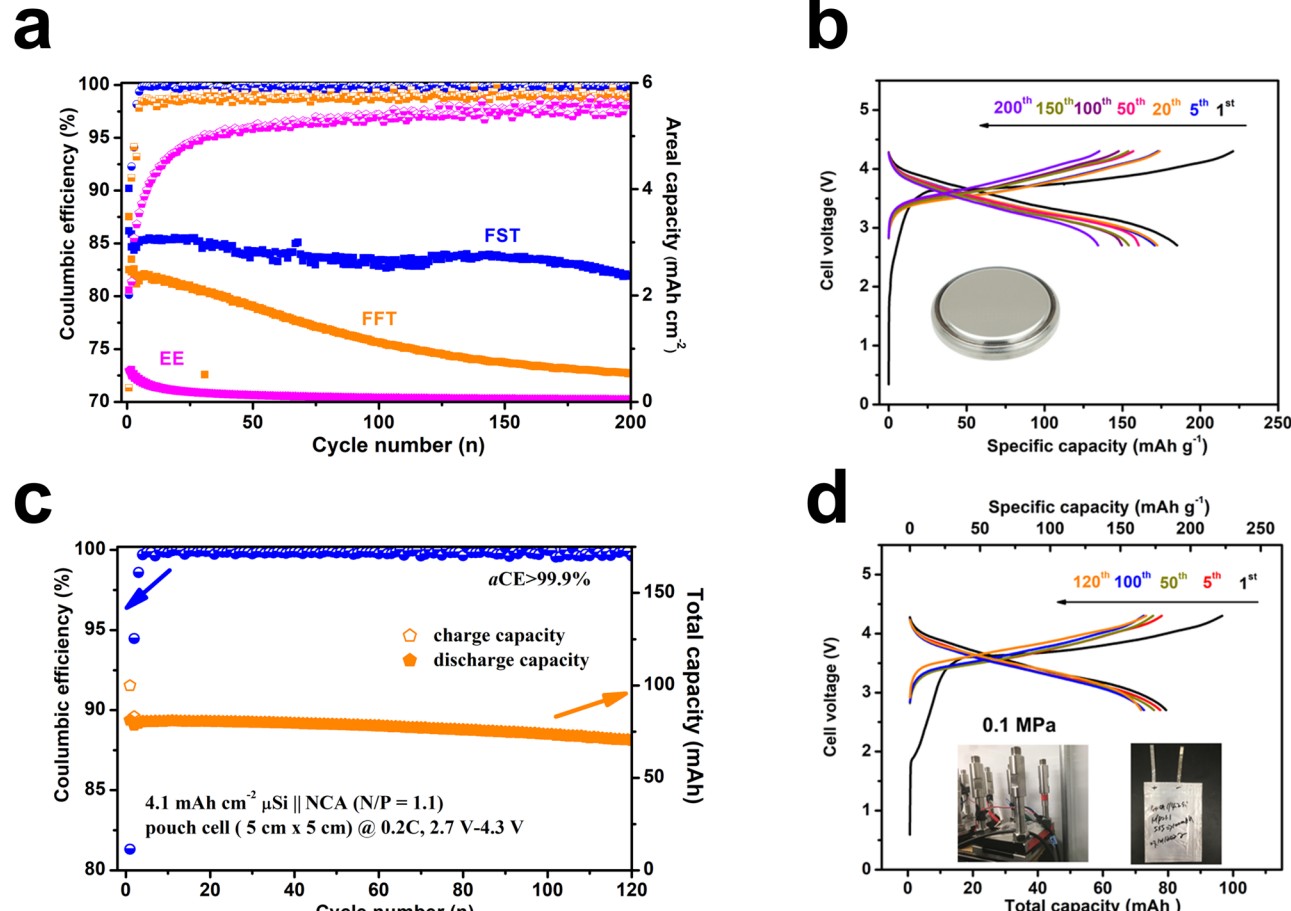

**Fig. 8 | Cycling of the μSi||NCA full cells. a** Long cycle performance of μSi||NCA (4 mAh cm⁻², N/P = 1.1) in coin cell configuration with comparison for the three investigated electrolytes. **b** Typical charge/discharge profiles of the μSi||NCA coin cell in FST electrolytes, the inset shows the assembled coin cell with CR2032 cell case. **c** Practical 100 mAh pouch cell (electrode size of 5 cm by 5 cm) performance with μSi anode at room temperature. **d** The charge/discharge profiles of the 100 mAh pouch cells at the 1$^{st}$, 5$^{th}$, 50$^{th}$, 100$^{th}$, and 120$^{th}$ cycle. The left inset figure illustrates the test conditions of the assembled pouch cell under the normal pressure of 0.1 MPa, and the right inset shows the actual cell size of 5 cm by 5 cm. For all cells, the cycle rate is C/5 at room temperature with the first formation cycle at C/20. Source data are provided as a Source Data file.

compared to FFT electrolyte ensure thin CEI thickness (Fig. 9a, b) and high anti-oxidation stability. In addition, the broad shoulder of the P-O signal (~529–535 eV) in the FST electrolytes suggests the co-existence of the S-O species, which might come from the decomposition of SL molecules.

In summary, we designed the nonflammable 1.0 M LiPF$_6$–FEC-SL-TTE (FST) electrolytes that combine PF$_6^-$ anion reduction and fluorinated solvent reduction to form LiF as well as SL reduction to form Li$_2$O and Li$_2$SO$_x$ SEI on silicon microparticles. The LiF-Li$_2$O SEI in FST electrolytes enabled SiMP electrodes at 4.1 mAh cm⁻² to provide a high-capacity release of >2700 mAh g⁻¹ for over 250 cycles with an initial CE of 85.6% and a cycling CE of >99.8%. The μSi||NCA full cells (>4.0 mAh cm⁻², and N/P ratio of 1.1) in FST electrolytes exhibited a long cycle life of >200 with a high cycling CE of 99.9% at a capacity of >4.0 mAh cm⁻². The practical 100 mAh large μSi||NCA pouch cells with >4.0 mAh cm⁻², and N/P ratio of 1.1 also demonstrated a stable (>120 cycles) and promising performance with high cycling CE of >99.9%. The electrolyte design by forming Li$_2$O SEI from solvent reduction opens new doors for next-generation high-energy Li-ion batteries, providing an alternative way other than the traditional thoughts of suppressing the reduction of solvents in the electrolytes. In addition, our proposed nonflammable FST electrolytes have the potential to commercialize the SiMPs pairing with market-available cathodes such as NCA.

## Methods
### General materials

Lithium hexafluorophosphate (LiPF$_6$, >99.99%) salt was purchased from Gotion, and Li chips with a thickness of 250 μm were purchased from MTI Corporation. The reference electrolytes 1.0 M LiPF$_6$ in EC/EMC = 50/50 (v/v) (battery grade) and fluoroethylene carbonate (FEC, 99%) were bought from Sigma-Aldrich. Methyl (2, 2, 2-trifluoroethyl) carbonate (FEMC, >98%), 1, 1, 2, 2-tetrafluoroethyl 2, 2, 3, 3-tetrafluoropropyl ether (TTE, >97%) and tetramethylene sulfone (SL, >99%) were purchased from TCI, US. All the solvents were dried over activated molecular sieves (4 Å, Sigma-Aldrich) to make sure the water content was less than 10 ppm (Karl-Fisher titrator, Metrohm 899 Coulometer). The LiNi$_{0.8}$Co$_{0.15}$Al$_{0.05}$O$_2$ (NCA) cathodes coated on Al foil with a loading of 4.0 mAh cm⁻² were kindly provided by Saft America, Inc. For the SiMP electrodes, a slurry was first prepared by dispersing SiMPs (1–5 μm, TCI, US, as-received, as revealed by SEM in Supplementary Fig. 23), lithium polyacrylate binder (10 wt% aqueous solutions) and Ketjen black in water with a weight ratio of 6:2:2. The slurry was cast onto a copper (Cu) foil, dried at room temperature for 24 h and further dried at 90 °C overnight under vacuum. μSi electrodes with a loading of 1.2 mg cm⁻² (corresponding to 4.3 mAh cm⁻² from a theoretical value of 3579 mAh g$_{Si}^{-1}$) were obtained. The μSi electrode processing is the same as that of commercial graphite electrodes without any additional pretreatment or pre-lithiation.

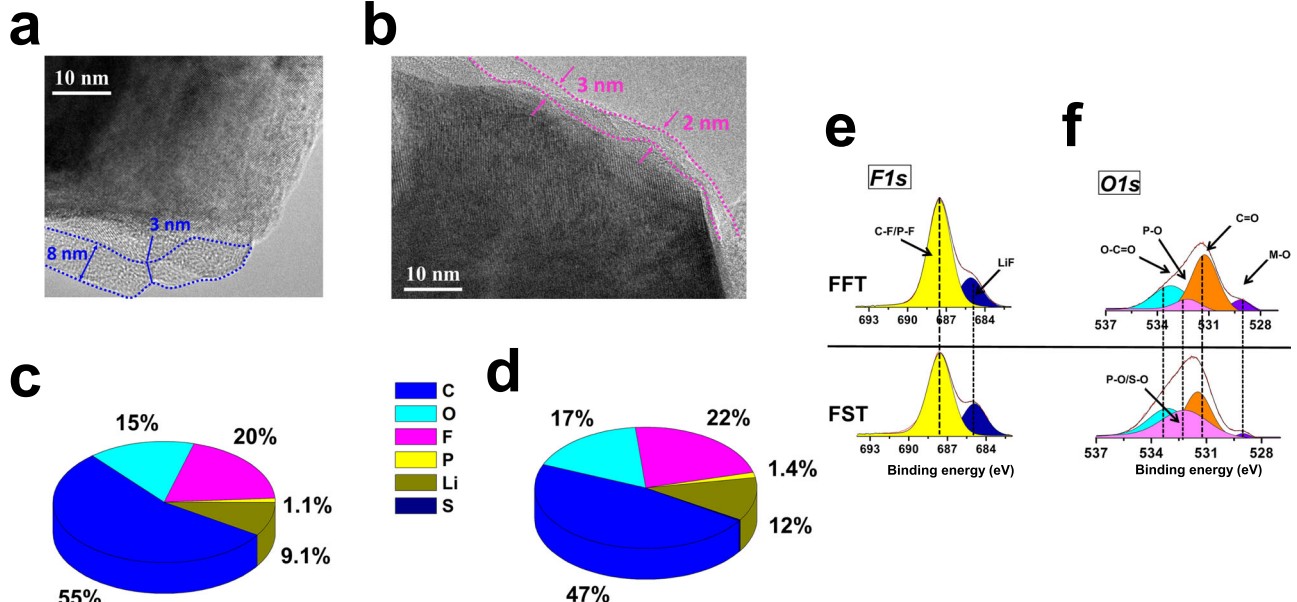

**Fig. 9 | Characterizations of cycled NCA electrodes in FFT and FST electrolytes. a–f** Typical HR-TEM images of cycled NCA electrodes recovered from µSi||NCA full cells after 50 cycles with FFT (**a**) and FST (**b**) electrolytes; the atomic distribution of CEI element on the surface of the NCA electrodes cycled in different electrolytes, FFT (**c**), FST (**d**); XPS *F1s* (**e**) and *O1s* (**f**) surface spectra for NCA cathodes with FFT (upper) and FST (below) electrolytes, The relative intensity for all spectra was shown in arbitrary units (a.u.) without labeling the *y*-axis for clarity.

## Electrolyte preparation

The reference electrolyte EE [1.0 M LiPF$_6$ in EC/EMC = 50/50 (v/v) (battery grade, Sigma)] was used as received, and the fluorinated electrolytes were prepared by first mixing the pure solvents FEC, FEMC, and TTE with a volume ratio of 2:6:2, then 1.0 M LiPF$_6$ was dissolved in the obtained mixture to get the FFT electrolytes. To prepare the FST electrolytes, a homogeneous solution of FEC, SL, and TTE by the volume ratio of 2:6:2 was first obtained by mixing the corresponding solvents. Then 1.0 M LiPF$_6$ was dissolved in the prepared mixture to get the FST electrolytes. The molarities here were calculated based on the moles of salt added and the volumes of solvents used. The ionic conductivities of the electrolytes were calculated by electrochemical impedance spectroscopy measurements with two platinum plate electrodes (1 ×1 cm$^2$) symmetrically placed in the electrolyte solutions.

## Electrochemical measurements

CR2032 coin-type half cells were assembled by sandwiching one piece of Celgard 3501/2325 separator between the SiMP electrodes and Li metal foil. The electrolytes used for cell assembly were: (1) EE [1.0 M LiPF$_6$ in EC/EMC = 50/50 (v/v)]; (2) FFT [1.0 M LiPF$_6$ in FEC/FEMC/TTE = 20/60/20 (v/v/v)]; and (3) FST [1.0 M LiPF$_6$ in FEC/SL/TTE = 20/60/20 (v/v/v)].

In the galvanostatic Li||µSi cell tests, the current density was set at 0.25 C (1C = 3579 mAh g$^{-1}$) in the potential range 0.05–1.0 V vs Li/Li$^+$ using a battery cycler (Landt Instrument), which is placed and operated in an open environment without temperature control (between 20 °C to 25 °C). For all electrolytes, one activation cycle with a voltage cutoff of 0.005 V at C/20 was performed before the cycling test at C/4. Both the specific capacities and current densities are based on the SiMP mass only. Linear sweep voltammetry (LSV) and cyclic voltammetry (CV) with different scan rates or voltage ranges were conducted on a CHI 600E electrochemical workstation (CH Instruments Inc. USA). The $^{19}$F-, and $^7$Li-NMR spectra were recorded on a Varian Mercury 400 MHz NMR spectrometer at room temperature. The Horiba Jobin Yvon Labram Aramis with a 532 nm diode-pumped solid-state laser was used for Raman measurements. Li$^+$ transference number (LTN), and

electrochemical impedance spectroscopy (EIS) were tested on a Gamry 1000E electrochemical workstation (USA) The electrochemical impedance spectroscopy measurements were taken over a frequency range of 1 MHz to 0.1 Hz. The transference number t$^+$ was calculated by the following Eq. (1):

$$t_+ = \frac{I_s(\triangle V - I_0 R_0)}{I_0(\triangle V - I_s R_s)} \tag{1}$$

where ΔV is the voltage polarization applied, I$_s$ and R$_s$ are the steady-state current and resistance, I$_0$ and R$_0$ are the initial current and resistance, respectively. The applied voltage bias for the LTN tests in the Li||Li cells here was 10 mV.

For SEM imaging of the electrodes after cycling, the electrodes were washed with corresponding mother liquor (without adding salt) to remove any residual Li salts from the surface of the electrodes and vacuum-dried before the sample was transferred to Hitachi SU-70 field emission gun SEM or a JEOL 2100 F field emission scanning transmission electron microscope (STEM) equipped with energy-dispersive spectroscopy (EDS, Bruker X-Flash 6/60 series) and Gatan image filter (GIF, Tridien 863) operating at 200 kV for the morphologies characterization. For STEM spectrum imaging (STEM-SI), we used a large dimension (~20 nm x20 nm) as a scanned pixel, and the acquisition time for each scanned pixel is 50 ms to explore the overall information. The ToF-SIMS attached with a Ga$^+$ focused ion beam (FIB)/SEM (Tescan GAIA3) was employed to do the ion sputtering.

For full cell tests, NCA cathodes coated on Al foil (4.0 mAh cm$^{-2}$) were kindly provided by Saft America Inc. The µSi||NCAfull cells (N/P of ~1.1) were charged/discharged between 4.3 V and 2.7 V in an open environment without temperature control (between 20 °C to 25 °C). The cells were cycled at C/5 (0.8 mA cm$^{-2}$) before one formation cycle at C/20 (0.2 mA cm$^{-2}$) without any pre-activation of the µSi electrodes. The 100 mAh homemade pouch cell is fabricated inside a glovebox, where aluminum and nickel strips are attached as electrode tabs to the sides of the cathode and anode, respectively. The electrolyte addition for each pouch cell was 3 g Ah$^{-1}$. The electrolyte was dropped into the package through a pipette, followed by the sealing of the battery under

vacuuming. The large pouch cell was cycled between 2.7 and 4.3 V on an Arbin battery test station (BT2000, Arbin Instruments) that is stored in a 25 °C testing room.

For XPS tests, data were collected using the **Kα** X-ray Photoelectron Spectrometer System (Thermo Scientific™, Al **Kα** radiation, **hν** = 1486.68 eV) at the University of Rhode Island. The sample preparation is the same as the SEM test. The sample was directly moved from the Ar atmosphere to the XPS chamber with a vacuum transfer container to avoid exposure to the air. The neutralizer was applied during the data collection, and an Ar sputter gun was used for the etching with the ion energy set at 200 eV and the middle range current selected. The sputtering rate was estimated to be -0.01 nm s⁻¹. The etching procedure was carried out in a cycle of accumulated 0, 60, 120, 180, 300, and 600 seconds. Spectra were recorded of the sample surface before sputtering and between sputtering cycles. All data was calibrated based on the *C1s* peak to 284.8 eV for binding energy values. Peak fitting and relative atomic percentage estimation were done using CasaXPS software (version 2.3.24)[47], after accounting for the relative sensitivity factors (R.S.F) of Thermo K-Alpha.

For PDF measurements. Electrolyte solvent, salt, and electrolyte solution were packed inside polyimide capillary tubes sealed by epoxy glue on both sides. The PDF measurements were carried out at the 28-ID-2 beamline of National Synchrotron Light Source II (NSLS II) In Brookhaven National Laboratory (BNL) using a photon wavelength of 0.1818 Å. The obtained data were integrated using Fit2D software[48]. The PDF and G(r) values were extracted using PDFgetX3 software.

## Calculation on the Li$_x$Si–LiF interface energy

First-principles calculations based on density functional theory (DFT) were performed using the Vienna ab initio simulation package code (VASP 6.3.0)[49]. The projector augmented wave (PAW) potentials were adopted and Perdew–Burke–Ernzerhof (PBE) realization of the generalized gradient approximation (GGA) for the exchange-correlation was performed[50,51]. The geometry optimizations were performed using the conjugated gradient method and the convergence threshold was set to be $10^{-5}$ eV in energy and $10^{-4}$ eV Å$^{-1}$ in force. The cutoff energy for the plane-wave-basis was 520 eV. Monkhorst–Pack **k**-point sampling was used for Brillouin zone integration. The crystal structures of Li$_x$Si and Li$_y$X (Li$_y$X = LiF and Li$_2$O) were obtained from the Material Projects database[52] and fully relaxed before use (ISIF = 3). The amorphous structure of Li$_x$Si was generated from Ab initio molecular dynamics (AIMD) simulations with a **Γ**-centered **k**-point. The relaxed crystal structures were first melted at a high temperature with NVT ensemble[53] for 2 ps and then rapidly quenched to 300 K at a rate of 1 K fs⁻¹. The annealing temperature is 1500 K to accelerate the melting process. The timestep is 2 fs. To build the surface slabs, a LiF supercell ($2 \times 2 \times 3$), Li$_2$O supercell ($2 \times 2 \times 2$) and Li$_2$CO$_3$ crystal ($1 \times 1 \times 1$) were cleaved along the (001), (111) and (110) surface respectively, which are surfaces with lowest surface energy among the low miller index planes. The work of separation for the Li$_x$Si– Li$_y$X interface is defined by Eq. (2):

$$W_{sep} = \frac{E_{LixSi} + E_{LiyX} - E_{LixSi-LiyX}}{A} \quad (2)$$

Where E$_{LixSi}$, E$_{LiyX}$ and E$_{LixSi-LiyX}$ are the total energy of the Li$_x$Si slab, Li$_y$X slab, and Li$_x$Si– Li$_y$X interface. A is the cross-sectional area of the interface slab.

## MD simulation methodology

Structural and transport properties of FST electrolytes were extracted from MD simulations employing a polarizable APPLE&P force field[54,55]. FF using the previously developed LiPF$_6$, FEC, and TTE parameters[56,57]. The SL FF parameters were modified as follows: charges were refit to electrostatic potential on a grid surrounding a molecule, its dipole and

quadrupole moments using both the optimized isolated SL geometry and the geometry from an SL/Li⁺ complex calculated using M05-2X/aug-cc-pvTz DFT. Electrostatic potential was calculated at MP2/aug-cc-pvTz level. Polarizability of the -SO$_2$ group of SL was reduced to prevent over-polarization in MD simulations resulting in 20% smaller molecular polarizability in FF compared to wB97XD/aut-cc-pvTz DFT. Molecular mechanics (MM) optimization using developed force field MM(FF) predicted the Li⁺(SL) and Li⁺(SL)$_2$ binding energies of −48.9 kcal mol⁻¹ and −82.7 kcal mol⁻¹, respectively, in good agreement with G4MP2 QC values of −49.4 kcal mol⁻¹ and −86.4 kcal mol⁻¹, respectively. Compared to the Li⁺(SL) binding energy in gas-phase, the Li⁺(FEC) binding energy is 6 kcal mol⁻¹ weaker −42.9 kcal mol⁻¹ from G4MP2 QC calculations and −41.3 kcal mol⁻¹ from MM(FF). An archive containing a final configuration from MD simulations, force field and simulation parameters is attached to the manuscript. A detailed description of the MD file formats and the associated MD simulation code was previously published by Borodin et al.[54] and is also included in the attached archive.

MD cells of FST contained 100 LiPF$_6$, 630 SL, 280 FEC, and 93 TTE, while MD cells of pure SL and FEC contained 512 solvents, and the TTE simulation box contained 216 molecules. Three independent replicas of the FST electrolytes were simulated at 90 °C and 60 °C. After 50 ns of simulations at 60 °C, 2 replicas were simulated at 25 °C. The length of total production runs, simulation temperatures, Li⁺ coordination numbers, and transport properties are summarized in Supplementary Table 2. The equations of motions were solved with a time reversible (RESPA) integrator with the following time steps: i) the contribution from bonds and angles to the forces were calculated at any 0.5 femtoseconds (fs), ii) the contribution of dihedrals and non-bonded forces within 8 Å cut-off was updated at any 1.5 fs, and iii) the remainder of the forces (reciprocal space Ewald using **k** = 8³ vectors and non-bonded forces within 15 Å cut-off was updated at any 3 fs. Nose-Hoover thermostat with 3 chains was used for temperature control with the associated frequency of 0.01 fs⁻¹. The induced dipoles (μ) were found self-consistently at each 3 fs timestep with the tolerance of $\mu^2 < 10^{-14}$ (e*Å)², where e is an electro charge.

We followed the previously published procedures for extracting transport properties from MD simulations[58]. Solvent and ion self-diffusion coefficients were extracted using the Einstein relation from linear fits to mean-square displacements divided by six in the diffusive regime. Due to the finite size of the simulation cells, long-range hydrodynamic interactions restrict the diffusion result in slowing down of ion and solvent diffusion. The leading order finite size correction (FSC) to the self-diffusion coefficient is given by Eq. (3):

$$\Delta D^{FSC} = \frac{2.837 k_B T}{6\pi\eta L} \quad (3)$$

where **k$_B$** is the Boltzmann constant, **T** is temperature, **L** is a linear dimension of the simulation periodic cell, and **η** is viscosity. Solvent and ion diffusion coefficients were corrected for the finite size using Eq. (3). The magnitude of correction is between 11% and 17%. Viscosity was calculated using the Einstein relation including both diagonal and non-diagonal elements to enhance the statistics using Eqs. (4)–(6) as results were shown to agree well with non-equilibrium methods:[59]

$$\eta = \lim_{t\to\infty} \eta(t) = \lim_{t\to\infty} \frac{V}{20 k_B T t} \left( \left\langle \sum_{\alpha,\beta} (L_{\alpha\beta}(t) - L_{\alpha\beta}(0))^2 \right\rangle \right) \quad (4)$$

$$L_{\alpha\beta}(t) = \int_0^t P_{\alpha\beta}(t\prime)dt\prime \quad (5)$$

where $k_B$ is the Boltzmann constant, $T$ is temperature, $t$ is time, $V$ is the volume of the simulation box, $P_{ab}$ is the stress sensor given by:

$$P_{\alpha\beta} = \frac{\sigma_{\alpha\beta} + \sigma_{\beta\alpha}}{2} - \frac{\delta_{\alpha\beta}}{3} tr(\sigma) \qquad (6)$$

where $\sigma_{ab}$ is the stress tensor with $\delta_{ab} = 1$ for $\alpha = \beta$ and $\delta_{ab} = 0$ for $\alpha \neq \beta$.

The degree of ion uncorrelated motion ($\alpha_d$) that is often called ionicity is around 0.6–0.7, indicating rather weak ionic correlations. It was extracted using Eqs. (7)–(9):

$$\alpha_d = \frac{\kappa}{\kappa_{uncorr.}} \qquad (7)$$

$$\kappa_{uncorr} = \frac{e^2}{V k_B T}(n_+ D_+ + n\_ D\_) \qquad (8)$$

$$\kappa = \lim_{t \to \infty} \frac{e^2}{6 t V k_B T} \sum_{i,j}^{N} z_i z_j \langle ([\mathbf{R}_i(t) - \mathbf{R}_i(0)]) ([\mathbf{R}_j(t) - \mathbf{R}_j(0)]) \rangle \qquad (9)$$

where $e$ is the electron charge, $V$ is the volume of the sample, $k_B$ is Boltzmann's constant, $T$ is the temperature, $n_+$ and $n_-$ are the number of cations and anions, respectively. Conductivity in Eq. (9) has contributions from the cation-cation, cation-anion, and anion-anion displacements denoted as $\sigma_{++}$, $\sigma_{+-}$ and $\sigma_{--}$. If one neglects ion correlations expressed by the off-diagonal elements in Eq. (9), conductivity $\kappa_{uncorr}$ with contributions only from ion self-diffusion is obtained (Eq. (8)).

The Li$^+$ cation transference number under anion-blocking conditions ($t_+^{abc}$) was extracted from MD simulations according to the methodology suggested by Roling' group[60]. It relies on the Onsager relations with linear response theory with additional assumptions The transference number ($t_+^{abc}$) depends on two parameters $\alpha$ (Roling), $\beta$ (Roling), where $\alpha$ yields cation contribution charge flux assuming no ion correlation, while $\beta$ accounts for the ionic correlations, see Eqs. (10)–(13):

$$\alpha = \frac{\sigma_{++}}{\sigma_{++} + \sigma_-} \qquad (10)$$

$$\beta = \frac{2\sigma_{+-}}{\sigma_{++} + \sigma_-} \qquad (11)$$

$$\sigma = \sigma_{++} + \sigma_{--} - 2\sigma_{+-} \qquad (12)$$

$$t_+^{abc} = \frac{\beta^2 - 4\alpha + 4\alpha^2}{4(1 - \alpha)(\beta - 1)} \qquad (13)$$

MD simulations predicted $t_+^{abc}$ in the range of 0.59–0.67 for FST electrolytes, which is much higher than $t_+^{abc} = 0.43$ predicted from MD simulations for 1 M LiPF$_6$ in EC-DMC(1:1 v/v). The higher value for $t_+^{abc}$ for FST is due to higher $\alpha$ of 0.44–0.50 (FSE) vs 0.39 for (1 M LiPF$_6$ in EC-DMC) and higher $\beta$ of 0.35–0.40 (FSE) vs 0.1 for (1 M LiPF$_6$ in EC-DMC). Higher $t_+^{abc}$ is consistent with the previously discussed ability of SL molecules to rotate and allow the exchange of Li$^+$ between solvent molecules, thus shifting the Li$^+$ diffusion mechanism from the vehicular-based one towards the structural diffusion[31].

### Density functional theory calculations

The Li$^+$(solvent) binding energies were calculated using DFT with the wB97XD functional, 6–31 + G(d,p) basis set with both solvent and Li$^+$(solvate) immersed in an implicit solvent that is modeled using PCM(ether) with solvent-excluded surfaces (Surface = SES keyword)

and without it PCM*(ether). The solvation model based on density with a higher dielectric constant $\varepsilon = 20$ SMD($\varepsilon = 20$) was also used as implemented in the Gaussian 16 software package, revision C.01.

The reduction potential for the complex A denoting either an isolated solvent or a solvate was calculated as the negative of the free energy of formation of A$^-$ in solution [$\Delta G^S_{298} = G^S_{298}(A^-) - G^S_{298}(A)$] divided by Faraday's constant as given by Eq. (14):

$$G^{red} = - \frac{\triangle G^S_{298K}}{F} - 1.4V \qquad (14)$$

The difference between the Li/Li$^+$ and the absolute reduction potential of 1.4 V was subtracted to convert results to the Li/Li$^+$ scale as discussed extensively elsewhere. Because both FEC and SL coordinate Li$^+$ their lithium solvates were used for calculating reduction potentials, while isolated TTE was used for the calculation of reduction potential because it was observed not to coordinate Li$^+$ in MD simulations.

## Data availability

The authors declare that the data supporting the findings of this study are available within the article and its Supplementary Information files. The data that support the plots within this paper are available from the corresponding author upon reasonable request. An archive containing a final configuration from MD simulations, force field and simulation parameters is provided as the supplementary file. Source data are provided with this paper.

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

## Acknowledgements

This work was supported by the U.S. Department of Energy (DOE) under award number DEEE0009183 at the University of Maryland (UMD) and via Interagency Agreement 89243322SEE000018 at DEVCOM ARL (C.S.W). The work at Brookhaven National Laboratory is supported by the Assistant Secretary for Energy Efficiency and Renewable Energy, Vehicle Technology Office of the US Department of Energy (DOE) through the Advanced Battery Materials Research (BMR) Program under contract no. DE-SC0012704 (E.Y.H. & X.Q.Y.). This research used a 28-ID-2 beamline of the National Synchrotron Light Source II, U.S. DOE Office of Science User Facilities, operated for

the DOE Office of Science by Brookhaven National Laboratory under contract no. DE-SC0012704 (E.Y.H. & X.Q.Y.).

## Author contributions

A.L. and C.W. conceived the idea for the project and wrote the manuscript. T.P. and O.B. helped with the molecular modeling and manuscript revision. Z.W. helped with the DFT modeling. S.L. and J.R. helped with the STEM measurement. T.L. helped with the Raman data collection. C.J. and B.L. helped with the XPS data collection. S.T., E.H., and X.Y. helped with the synchrotron data collection and analysis. W.Z. and all other authors discussed the results and analyzed the data.

## Competing interests

The authors declare no competing interests.
