## [Peer Review File · Nature Communications]

High Voltage Electrolytes for Lithium-Ion Batteries with Micro-Sized Silicon AnodesREVIEWER COMMENTS

Reviewer #1 (Remarks to the Author):

The authors demonstrated high voltage electrolytes for Li-ion batteries with micron-sized Si anodes. Typically, the pulverization of micron-sized Si and solid-electrolyte-interphase (SEI) layers allow electrolytes to penetrate the cracked micron-Si and form new SEI, which isolates the pulverized micron-Si resulting in a rapid capacity decay. In this work, the authors developed a novel electrolyte that forms a Si-phobic Li₂O-LiF SEI with weak bonding to micron-Si particles which can achieve high Coulombic efficiency for both micron-Si and high-voltage NCA cathode. The resulting electrochemical performance is much better than those of previous numerous reports. This strategy is valuable broad interest to be reported in Nature Communications. However, several raised issues should be described.

1. Why is the Li₂O Si-phobic material? The authors should describe the more detail for this point.
2. In the Figure 6d, LiF and Li₂O are alternatively placed in parallel. Is there any evidence for this placement?
3. In the Figure 7d, areal capacity of SiMPs was significantly decayed at early stage. The authors should describe more clearly this result.
4. The authors showed the SiMPs expansion trend in the Figure 9. After 50 cycles, 2 times higher electrode swelling was observed compared to the pristine electrode. Is there any problem to operate the pouch full cell without additional pressure? It may cause the contact loss between anode and cathode.
5. In the Figure 9, after 50 cycles, the authors showed the FIB cross-sectional SEM images. Micrometer-sized Si particles in the FFT and EE electrolytes seem to be merged after pulverization and the formation of a large amount of SEI layers. In contrast, Si particles in the FST electrolytes retained spherical morphology at certain local area. The other parts look like a serious agglomeration that is similar to other electrolyte system. More clear evidence is required.
6. In page 4, FEC-FEMC-TTE is the first word. It should be written with full name.
7. In page 5, "1,1,2,2-Tetrafluoroethyl-2,2,3,3-Tetrafluoropropyl Ether" should be corrected as "1,1,2,2-tetrafluoroethyl-2,2,3,3-tetrafluoropropyl ether".

Reviewer #2 (Remarks to the Author):

Micro-sized Si anodes have a higher capacity than graphite and a lower cost and longer calendar life compared to the nano-sized ones. However, the μ Si anode normally shows a poor cycle life due to Si particle fracture and SEI broken originating from the large volume change during lithiation/de-lithiation. The authors reported a new concept of forming Si-phobic LiF to stabilize micro-Si by using LiPF₆/mixed THF electrolytes. However, the low anodic stability and low boiling point of mixed THF solvents impede the practical applications of LiPF₆/mixed THF electrolytes. The authors now reported Si-phobic Li₂O-LiF SEI by using high-voltage carbonate-based electrolytes, allowing the μ Si/NCA at high areal loadings of >4 mAh cm⁻² and low N/P ratio of ~1.1 to achieve 200 cycles. This work represents the first electrolyte that allows the high-voltage LIBs with μ Si anode. It is highly recommended for publication in the Journal of Nature Communication. A few comments for the authors to consider before acceptance:

1. The XPS was used to study the SEI components and distributions. However, some detailed information should be included in the experimental part. Is a neutralizer applied?

Which sputtering gun was used for the etching, at which current and voltage? These are important parameters to consider when analyzing the XPS data. The contents and ratios of Li₂O and LiF in SEI for EE, FFT and FST electrolytes should be provided and discussed to support the SEI variations among these electrolytes.

2. Figure 7 depicts the electrochemical performance of the 5 μm silicon electrode in Li/μSi half cells. However, the voltage profiles in Figure 7a-c are not well displayed. The reviewer suggests removing the areal capacity but highlighting the specific capacity of the μSi electrode in these figures. In addition, the Li/μSi half cells are cycled between 0.05-1V, why the cut-off voltage is not selected from 0.01-1V?

3. The authors showed that the μSi cycled in FST electrolyte “have a “crack-less” morphology and Si particles larger than 5 μm could still be noticed after 50 cycles” and no micro-sized particles were observed in the FIB cross-section of the electrodes for the reference electrolytes. Theoretically, μSi particle cracks will be inevitable during lithiation/delithiation, why the micro-sized morphology is retained in FST electrolytes? Moreover, why does μSi electrode swell largely in the first few cycles and is restrained after 20 cycles for FST electrolytes, but not the reference electrolytes?

4. Minor issues: 1) Figure labelings: In Figure 6, the labelings should be unified with the same font and text size; in Figures 8 and 9, the resolution labelings are too small to be recognized, they should be enlarged with better resolution; 2) The electrolyte flammability test should be provided with recorded videos, a single image can not tell the full examination of the investigation; 3) In the full-cell data of Fig.10d, why does the capacity fluctuate during cycles?

Lastly, 12 figures are exceptional for an article, the authors are suggested to combine the figures or move some figures to the supporting information.

Point-by-Point Response to Referee's Comments (Manuscript ID: NCOMMS-23-39016)

We very much appreciate the reviewers for their high recommendation and constructive comments on our manuscript. During the past few months, we have conducted more experiments and made careful revisions to our manuscript to address all the concerns raised by the reviewers (listed below). In the following, we provide point-by-point replies to the reviewer's comments. The main text as well as the supplemental materials have been revised accordingly and highlighted in yellow.

New experiments and discussions:

- Li_2O as Si-phobic material has been validated through additional **DFT calculations**. We also calculated the Work of Separation (WoS) value for Li_2CO_3 and LiF as SEI component reference. The corresponding discussion was added in the main text and **Supplementary Note 4** to address the phobicity of Li_2O as the SEI component on the silicon anode as suggested by **Reviewer #1**.
- Parallel pouch full cells of $\mu\text{Si}||\text{NCA}$ (4 mAh cm^{-2} , N/P = 1.1) have been assembled and tested under different **external pressures**. The pressure-dependent pouch cell performance and the corresponding discussion have been added to the revised manuscript. Minimum pressure is required for the $\mu\text{Si}||\text{NCA}$ pouch cells, which may be the same for all pouch cells to maintain good electrical contact.
- The **post-cycle evaluation of SiMP electrodes** after 200 cycles in different electrolytes has been conducted. Scanning electron microscope (**SEM**) and energy-dispersive X-ray (**EDX**) results showed a uniform Li_2O -LiF SEI formation on the SiMPs after the extended cycles, even SiMPs cracks, allowing the μSi anode to have superior cycle performance with restrained SiMPs swelling and electrode volume expansion.
- **Details for the XPS data collection** have been provided as requested by **Reviewer #2**, and a complete analysis has been given and added in the revised manuscript to show the Li_2O formation and distribution in the designed FST electrolytes.
- Following **Reviewer #2's** suggestion, we have carefully unified the labels on all figures, moved three figures to the Supporting Information, and merged Figures 3 and 4, Figures 10 and 11, with nine primary figures shown in the revised manuscript.
- **7 Figures** have been added/moved to the Supporting Information.
- **3 recorded videos** were added as supplementary materials requested by **Reviewer #2**.

REVIEWER COMMENTS

Reviewer #1 (Remarks to the Author):

The authors demonstrated high-voltage electrolytes for Li-ion batteries with micron-sized Si anodes. Typically, the pulverization of micron-sized Si and solid-electrolyte-interphase (SEI) layers allow electrolytes to penetrate the cracked micron-Si and form new SEI, which isolates the pulverized micron-Si resulting in a rapid capacity decay. In this work, the authors developed a novel electrolyte that forms a Si-phobic Li₂O-LiF SEI with weak bonding to micron-Si particles which can achieve high Coulombic efficiency for both micron-Si and high-voltage NCA cathode. The resulting electrochemical performance is much better than those of previous numerous reports. This strategy is valuable broad interest to be reported in Nature Communications. However, several raised issues should be described.

Response: We would like to express our gratitude to the reviewer for his/her careful evaluation of our manuscript and insightful comments. We appreciate the reviewer's endorsement of our work and have addressed the reviewers' issues in the point-to-point response as follows.

1. Why is the Li₂O Si-phobic material? The authors should describe the more detail for this point.

Response: We thank the reviewer for the comment. The “phobicity” of one compound towards a specific material can be reflected by the interfacial energy between them. If the interfacial energy is high, then a low adhesion between the compound and substrate is expected. Therefore, the Work of Separation (WoS) for different SEI components to Li_xSi phases can be calculated and correlated to the adhesion strength, which will correlate to their corresponding Si-phobicity. To evaluate the Si-phobicity of LiF and Li₂O to silicon/Li_xSi, we added Li₂CO₃ as a reference, which is also an important SEI component in organic electrolytes. The calculated WoS values for LiF, Li₂O, and Li₂CO₃ with different lithiated silicon (from Li₁₅Si₄, Li₁₂Si₇ to LiSi) are shown in **Figure R1**, where a low WoS indicates a high interface energy. The WoS values for LiF|Li_xSi, Li₂O|Li_xSi, and Li₂CO₃|Li_xSi interfaces are in the range of 0.13-0.27, 0.26-0.33 and 0.50-1.10 J m⁻², respectively, indicating much higher interfacial energy of LiF and Li₂O than that of Li₂CO₃ against Li_xSi phase.

Figure R1. The work of separation for LiF|Li_xSi, Li₂O|Li_xSi, and Li₂CO₃|Li_xSi interfaces.

The electron localization function (ELF) images for LiF|Li_xSi, Li₂O|Li_xSi, and Li₂CO₃|Li_xSi interfaces are shown in **Figure R2–R4** with the boundary between Li_xSi and Li_yX (X = F, O & CO₃) marked with red lines. A region with an ELF value of <0.2 was observed for LiF|Li_xSi and Li₂O|Li_xSi interfaces, representing the absence of bonding between interfacial atoms. In contrast, the ELF value between Li₂CO₃|Li_xSi interfaces varies from 0 to 0.9, corresponding to the formation of mixed ionic and covalent bonds. Based on the above data and analysis, the Li₂CO₃|Li_xSi has much lower interfacial energy than that of LiF|Li_xSi, Li₂O|Li_xSi interfaces, and the same value range of both WoS and ELF for LiF|Li_xSi, Li₂O|Li_xSi further confirms that Li₂O is a high modulus materials to Li_xSi phases, which can be used as a Si-phobic component in the μSi SEI.

Figure R2. Electron localized function and interfacial energy (E_{int}) between LiF and different lithiated Si phases. (a) LiF|Li₁₅Si₄, (b) LiF|Li₁₂Si₇, (c) LiF|LiSi.

Figure R3. Electron localized function and interfacial energy (E_{int}) between Li₂O and different lithiated Si phases. (a) Li₂O|Li₁₅Si₄, (b) Li₂O|Li₁₂Si₇, (c) Li₂O|LiSi.

Figure R4. Electron localized function and interfacial energy (E_{int}) between Li_2CO_3 and different lithiated Si phases. (a) $\text{Li}_2\text{CO}_3|\text{Li}_{15}\text{Si}_4$, (b) $\text{Li}_2\text{CO}_3|\text{Li}_{12}\text{Si}_7$, (c) $\text{Li}_2\text{CO}_3|\text{LiSi}$.

Figures R1–R4 were added to **Figure 4** in the revised manuscript, the corresponding discussion has been supplemented in **Supplementary Note 4** and the revised manuscript (highlighted in yellow below).

The revised manuscript on pages 13–14:

Adhesion of SEI components to Li_xSi plays a critical role in stabilizing the SiMP anode. The adhesion of SEI components can be reflected by the Work of Separation (WoS). The WoS for Li_2O , LiF and Li_2CO_3 to Li_xSi was calculated using molecular modeling, where Li_2CO_3 is used as a reference. A low WoS value corresponds to a high interface energy (E_{int}) (**Supplementary Note 4**). **Fig. 4a** shows that LiF and Li_2O have lower WoS values ($< 0.33 \text{ J m}^{-2}$) to Li_xSi ($\text{Li}_{15}\text{Si}_4$, $\text{Li}_{12}\text{Si}_7$ and LiSi) than Li_2CO_3 (up to 1.10 J m^{-2}), indicating higher interfacial energies of LiF and Li_2O to the active silicon particles during lithiation process. Electron localization function (ELF) images of the three SEI components (LiF , Li_2O , and Li_2CO_3) to the lithiated silicon phases are shown in **Fig. 4b–d**. A region with an ELF value of < 0.2 was observed for $\text{LiF}|\text{Li}_x\text{Si}$ and $\text{Li}_2\text{O}|\text{Li}_x\text{Si}$ interfaces, referring to the low chemical bondings at the interface. In contrast, the ELF value at the $\text{Li}_2\text{CO}_3|\text{Li}_x\text{Si}$ interface varies from 0 to 0.9, corresponding to the formation of mixed ionic and

covalent bonds. The Li_2O and LiF have high E_{int} to Li_xSi , and the Si-phobic Li_2O - LiF SEI suffer less stress during the large volume change of SiMPs.

The following texts have been added as Supplementary Note 4 on page 7 of the revised supplementary information:

Supplementary Note 4

Discussion of electron localized function for the Li_xSi alloy-SEI components interfaces

In the Li_xSi region, the red basin with an ELF value of 0.8-1 was observed around the Si core while a flat green profile with an ELF value of around 0.3-0.5 dispersed (**Fig. 4b-d**, main text). With the increase of Si content (from $\text{Li}_{15}\text{Si}_4$ to $\text{Li}_{12}\text{Si}_7$ then to LiSi), the ELF value of the red basin increases, representing more covalent characteristics between Si-Si bonds. Meanwhile, the flat green region shrinks when Li content decreases due to fewer metallic bonds in the Li_xSi phase. ELF map in LiF and Li_2O region shows a sphere shape with an ELF value of ~ 0.8 , representing ionic bonds with charge completely transferred between ions. The bond basins were observed within the Li_2CO_3 region due to the covalency of C-O bonds. In the boundary of Li_xSi and Li_yX , a region with an ELF value of < 0.2 was observed for $\text{LiF}|\text{Li}_x\text{Si}$, $\text{Li}_2\text{O}|\text{Li}_x\text{Si}$ interfaces, indicating the absence of bonding between the atoms at the interface. In contrast, the ELF value between the $\text{Li}_2\text{CO}_3|\text{Li}_x\text{Si}$ interfaces varies from 0 to 0.9, corresponding to the formation of mixed ionic and covalent bonds. Therefore, the interfacial energy for $\text{Li}_2\text{O}|\text{Li}_x\text{Si}$ interfaces is comparable to $\text{LiF}|\text{Li}_x\text{Si}$ and is much higher than that of $\text{Li}_2\text{CO}_3|\text{Li}_x\text{Si}$ interfaces, validating the Si-phobic of Li_2O with weak bonding to Li_xSi , which will release the SEI stress during the volume change of SiMPs and improve the μSi anode cycle performance.

2. In the Figure 6d, LiF and Li_2O are alternatively placed in parallel. Is there any evidence for this placement?

Response: We thank the reviewer for the comment. **Figure 6d** of the original submission is the cartoon demonstration for the DFT calculation model. This simplified model was set with alternatively distributed LiF and Li_2O to allow the approximation of both components on the conductivity effect. To clear any further misunderstanding, we have moved this figure to the

supporting information as **Supplementary Figure 16a** and added additional comments to indicate that this is only the demonstration of the simplified model in the revised manuscript (marked in yellow below).

The revised part on page 8 of the supporting information:

In this model, the topological distribution of the LiF and Li₂O phases was simplified as alternatively parallel so that the Li⁺ conduction path could penetrate along the SEI (**Supplementary Fig. 16a**). The simplified model provides an upper limit estimation of the ionic conductivity in actual SEI, where the tortuosity factor also affects Li⁺ conduction significantly.

The updated Supplementary Figure 16 in the revised supporting information with more captions added to indicate the simplified model demonstration:

Supplementary Figure 16. (a) Cartoon demonstration of the simplified model where the topological distribution of the LiF and Li₂O phases was set to be alternatively parallel so that the Li⁺ conduction path could penetrate along the SEI. (b) Formation energy v.s Fermi level (referenced to the VBM) for the most favorable native defects in LiF and Li₂O under lithium-rich ($\mu_{\text{Li}} = 0$) chemical potential conditions. Transitions are marked with stars. (c) The formation energy of Schottky defects was $E^f[\text{Schottky}_{\text{LiF}}^0] = E^f[\text{vac}_{\text{Li}}] + E^f[\text{vac}_{\text{F}}]$ and its evolution with Fermi energy was plotted by a dashed orange line. Based on defect formation energy calculation, the dominant point defects of LiF and Li₂O in equilibrium with the Li anode are Schottky defects and Li⁺ interstitial defects, respectively.

3. In the Figure 7d, areal capacity of SiMPs was significantly decayed at early stage. The authors should describe more clearly this result.

Response: We thank the reviewer for his/her careful observation of our data. The capacity decay of the SiMPs at the early stage (**Figure 5** in the revised version) is because we increased the current density from C/20 to C/4 and raised the cut-off voltage from 0.005 to 0.05V. To force the electrolytes to penetrate the μSi electrode for homogeneous lithiation, the formation cycle of $\text{Li}||\mu\text{Si}$ half cells at a low current of C/20 with a wide cut-off voltage of 0.005 V-1.0V was used. After the first formation cycle, the current density was increased to 0.25C and the cut-off voltage range was narrowed to 0.05–1.0 V *v.s* Li/Li^+ . The lithiation potential was improved to 0.05 V to reflect the actual N/P ratio of >1.1 in the full cell. The increased current density and lithiation cut-off potential after the first activation cycle led to the capacity drop in the second cycle. The FST electrolytes were able to deliver a high capacity of 2718 mAh g^{-1} after the formation cycle, indicating a high SiMPs capacity utilization.

We explained the capacity drop in the revised manuscript (marked in yellow below).

The revised part on page 16:

The μSi electrodes show a high initial capacity of 4.1 mAh cm^{-2} and $\sim 3,380 \text{ mAh g}^{-1}$ with initial Coulombic efficiency (*i*CE) of 85.6% in the formation cycle at a current density of 0.05 C, discharge cut-off potential of 0.005 V in the FST electrolytes (**Fig. 5a**). After the first cycle, the μSi electrodes were charged/discharged at a high current density of 0.25 C and high discharge cut-off potential of 0.05 V. The CE of the μSi electrode increases to 96.8% at the 2nd cycle and then to 99.3% in the 3rd cycle with an average Coulombic efficiency (*a*CE) of 99.8% from the 2nd to 250th cycle. The $5\mu\text{m Si}$ in FST electrolytes was able to deliver a high capacity of $\sim 2718 \text{ mAh g}^{-1}$ at 0.25 C with a capacity retention of over 80% after 250 cycles (**Fig. 5a, d**).

4. The authors showed the SiMPs expansion trend in the Figure 9. After 50 cycles, 2 times higher electrode swelling was observed compared to the pristine electrode. Is there any problem to operate the pouch full cell without additional pressure? It may cause the contact loss between anode and cathode.

Response: We appreciate the reviewer's comments. The soft pouch cells are assembled with all the cell components enclosed in an aluminum-coated plastic film. Low pressure will lead to the

loss of contact during the cycle of the cells. Per reviewer's request, we assembled three 2 cm x 2 cm pouches for the $\mu\text{Si}||\text{NCA}$ (4.0 mAh cm^{-2} , $\text{N/P} = 1.1$) full cells, and cycled these pouch cells using FST electrolytes under different conditions: (1) no pressure as the reviewer advised; (2) with external pressure (0.1 MPa) for 10 cycles, then remove the pressure; and (3) with external pressure (0.1 MPa) through the whole cycling.

Figure R5. The effect of external pressure on $\mu\text{Si}||\text{NCA}$ (4 mAh cm^{-2} , $\text{N/P} = 1.1$) pouch cell cycling. The μSi pouch cells were assembled with practical electrode loadings ($\sim 4 \text{ mAh cm}^{-2}$ for NCA and $\sim 4.1 \text{ mAh cm}^{-2}$ for μSi , electrode size of $\sim 2 \text{ cm}$ by 2 cm) and cycled with FST electrolytes. Before cycling at $\text{C}/5$, one formation cycle at $\text{C}/20$ was conducted. The external pressure was applied with a steel presser at 0.1 MPa.

All pouch cells were cycled at $\text{C}/5$ after one formation cycle at $\text{C}/20$. As shown in **Figure R5**, the application of external pressure indeed has a vital effect on the cycle performance of the $\mu\text{Si}||\text{NCA}$ pouch cells. Without compressing (**Figure R5**, black), the pouch cell showed inferior cycle stability with a low initial Coulombic efficiency (iCE) of 40.6% and the cell capacity dropped to $<50 \text{ mAh g}^{-1}$ in 10 cycles. This might be attributed to the inefficient wetting between the electrolyte/electrode and the possible loss of contact after the first few cycles. With the assistance of the external pressure, the pouch cell was able to obtain a high iCE of $>80\%$ for both cells (**Figure R5**, blue and magenta), and no capacity decrease was observed for the first 10 cycles of the cells. However, after the removal of the external pressure, the pouch cell showed a quick capacity and CE decay (**Figure R5**, blue). In sharp contrast, with the continuous application of 0.1 MPa external pressure, the pouch cell showed superior cycle performance with a high capacity retention of $>95\%$

and cycle CE of >99.9% in 100 cycles (**Figure R5**, magenta). This result demonstrated that the minimum external pressure is required to maintain good contact between the electrolytes and electrodes to achieve a long cycle life for $\mu\text{Si}||\text{NCA}$ pouch cells, which is in good agreement with the previous reports (*Electrochimica Acta*, 419,2022, 140354).

Figure R5 was added as **Supplementary Figure 33** and the above discussion as well as reference (*Electrochimica Acta*, 419,2022, 140354) has been updated in the revised manuscript (marked in yellow below).

The revised part on pages 23–24:

In addition, even though small, the 0.1 MPa external pressure has been proven to be essential for the successful cycle of the $\mu\text{Si}||\text{NCA}$ pouch cells, which could ensure good electrolyte/electrode contact during the cell cycling (**Supplementary Fig. 33**).⁴⁵

5. In the Figure 9, after 50 cycles, the authors showed the FIB cross-sectional SEM images. Micrometer-sized Si particles in the FFT and EE electrolytes seem to be merged after pulverization and the formation of a large amount of SEI layers. In contrast, Si particles in the FST electrolytes retained spherical morphology at certain local area. The other parts look like a serious agglomeration that is similar to other electrolyte system. More clear evidence is required.

Response: We thank the reviewer for the comments and constructive suggestions. We agree with the reviewer that the morphology difference of μSi particles after the 50 cycles in different electrolytes is not significant (Figure 9 in the original submission). As suggested by the reviewer, we further examined the Si particle evolution after 200 cycles using the scanning electron microscope (SEM) and energy-dispersive X-ray spectroscopy (EDX) (**Figure R6**), where the difference in capacity retention and electrode thickness of the μSi electrode in $\text{Li}||\mu\text{Si}$ (4.1 mAh cm^{-2}) cells are large in the three electrolytes (EE, FFT, and FST). As shown in **Figure R6a**, the SiMPs after 200 cycles in FST electrolytes showed less pulverization. The $\text{Li}_2\text{O-LiF}$ composite SEI formed in FST electrolytes was able to keep the integrity of SiMPs after the 200 repeated lithiation/delithiation cycles. Therefore, the large-sized SiMPs region was well-preserved with good contact with the binder and carbon black, which allows the reversible cycle of the μSi electrode with high capacity utilization. In addition, a homogeneous distribution of C, O, and F was identified in the elemental mapping, validating a uniform $\text{Li}_2\text{O-LiF}$ SEI layer formation

(**Figure R7**). However, large fractures and pulverized Si particles were found for SiMPs cycled in the reference electrolytes (EE and FFT) with porous morphology because the SEI in cracked Si further separates the pulverized Si (**Figure R6b–c**). The C, O, and F elements were found unevenly spreading over the electrode with a high intensity of C and O, correlating to the organic-rich SEI that leads to continuous electrolyte penetration and further SiMP pulverization (**Figure R8–9**). The large void in the swelled μ Si electrode leads to the loss of contact between active SiMPs and carbon black, resulting the fast capacity decay. This result is also consistent with the dramatic electrode thickness differences after 200 cycles in the three electrolytes (72 ± 1 , 113 ± 3 , and $47 \pm 2 \mu\text{m}$ for EE, FFT, and FST, respectively) (**Figure R6d**).

Figure R6. Morphology of Si particles and electrode thickness evolution after 200 long cycles. Focused ion beam (FIB) cross-section SEM images of the SiMP electrode after 200 cycles of operation in different electrolytes. **a–c**, **a**, FST. **b**, FFT. **c**, EE. **d**, The histogram of μ Si electrode thickness evolution in the three electrolytes.

Figure R7. Morphology and energy dispersive X-ray (EDX) mapping analysis of Si particles after 200 cycles in FST electrolytes. The C, O, F, and P signals are shown in yellow, blue, green, and magenta, respectively. The last photo shows the combination of C, O, F, and P mapping, representing the overview of SEI distribution on the SiMPs.

Figure R8. Morphology and energy dispersive X-ray (EDX) mapping analysis of Si particles after 200 cycles in FFT electrolytes. The C, O, F, and P signals are shown in yellow, blue, green, and magenta, respectively. The last photo shows the combination of C, O, F, and P mapping, representing the overview of SEI distribution on the SiMPs.

and magenta, respectively. The last photo shows the combination of C, O, F, and P mapping, representing the overview of SEI distribution on the SiMPs.

Figure R9. Morphology and energy dispersive X-ray (EDX) mapping analysis of Si particles after 200 cycles in EE electrolytes. The C, O, F, and P signals are shown in yellow, blue, green, and magenta, respectively. The last photo shows the combination of C, O, F, and P mapping, representing the overview of SEI distribution on the SiMPs.

Figure R6 was added to **Figure 7** in the revised manuscript and **Figures R7–R9** were added as **supplementary Figures 27–29**, and these results and discussions have been updated in the revised manuscript (marked in yellow below).

The revised part on pages 20–21 of the revised manuscript:

As shown in **Fig. 7a–c**, the SiMPs after charge/discharge in FST electrolytes for 200 cycles showed “crack-less” morphology (**Fig. 7a**), similar to the crack-free pristine Si with expanded size and deformed shape (**Supplementary Fig. 23b**). Only minor fractures were found in the SiMPs electrode. In addition, a homogeneous distribution of C, O, and F was identified in the elemental energy-dispersive X-ray spectroscopy (EDX) mapping (**Supplementary Fig. 27**), validating a uniform $\text{Li}_2\text{O-LiF}$ SEI layer formation. In sharp contrast, large fractures with huge porous

structures have developed in SiMPs cycled with the reference electrolytes (**Fig. 7b** for FFT, **7c** for EE). The C, O, and F elements were found unevenly spreading over the electrode with a high intensity of C and O for the SiMPs cycled with FFT and EE electrolytes (**Supplementary Figs. 28–29**), correlating to the organic-rich SEI that leads to continuous electrolyte penetration and further SiMP pulverization. The large pores in the swelled μSi electrode lead to the loss of contact between active SiMPs and carbon black, resulting the fast capacity decay.

6. In page 4, FEC-FEMC-TTE is the first word. It should be written with full name.

Response: We thank the reviewer for the comment. A full name for the FEC-FEMC-TTE was added in parentheses on page 4 of the revised manuscript when FEC-FEMC-TTE was first used (highlighted in yellow below).

The revised part on page 4:

For example, the high-voltage all-fluorinated carbonate electrolytes (1.0 M LiPF_6 in **FEC** (fluoroethylene carbonate)-**FEMC** (2,2,2-trifluoroethyl, methyl carbonate)-**TTE** (1,1,2,2-tetrafluoroethyl-2,2,3,3-tetrafluoropropyl ether) (denoted as FFT) enable μSi anode to achieve a CE of 99.7% when cycled at a low capacity of $>1000 \text{ mAh g}^{-1}$.²⁹

7. In page 5, “1,1,2,2-Tetrafluoroethyl-2,2,3,3-Tetrafluoropropyl Ether” should be corrected as “1,1,2,2-tetrafluoroethyl-2,2,3,3-tetrafluoropropyl ether”.

Response: We thank the reviewer for his/her careful observation. We apologize for the incorrect grammar errors and have corrected them in the revised manuscript (highlighted in yellow below).

The revised part on page 5:

Herein, we report a 4.3V carbonate electrolytes consisting of 1.0 M LiPF_6 salt and a 2:6:2 (by volume) mixture of fluoroethylene carbonate (FEC), sulfolane (SL), and 1,1,2,2-tetrafluoroethyl-2,2,3,3-tetrafluoropropyl ether (TTE) for $\mu\text{Si}||\text{NCA}$ cells.

Reviewer #2 (Remarks to the Author):

Micro-sized Si anodes have a higher capacity than graphite and a lower cost and longer calendar life compared to the nano-sized ones. However, the μSi anode normally shows a poor cycle life due to Si particle fracture and SEI broken originating from the large volume change during lithiation/de-lithiation. The authors reported a new concept of forming Si-phobic LiF to stabilize micro-Si by using LiPF₆/mixed THF electrolytes. However, the low anodic stability and low boiling point of mixed THF solvents impede the practical applications of LiPF₆/mixed THF electrolytes. The authors now reported Si-phobic Li₂O-LiF SEI by using high-voltage carbonate-based electrolytes, allowing the μSi /NCA at high areal loadings of $>4 \text{ mAh cm}^{-2}$ and low N/P ratio of ~ 1.1 to achieve 200 cycles. This work represents the first electrolyte that allows the high-voltage LIBs with μSi anode. It is highly recommended for publication in the Journal of Nature Communication. A few comments for the authors to consider before acceptance:

Response: We sincerely thank the reviewer's high rating and recommendation for publishing our work in the esteemed Journal of *Nature Communication*. The following is the point-by-point response to the reviewer's valuable and constructive suggestions.

1. The XPS was used to study the SEI components and distributions. However, some detailed information should be included in the experimental part. Is a neutralizer applied? Which sputtering gun was used for the etching, at which current and voltage? These are important parameters to consider when analyzing the XPS data. The contents and ratios of Li₂O and LiF in SEI for EE, FFT and FST electrolytes should be provided and discussed to support the SEI variations among these electrolytes.

Response: We thank the reviewer for his/her careful observation and the nice suggestion.

We provided the details for XPS data collecting in the following:

- 1) Yes, we applied the neutralizer during our XPS experiment.
- 2) The Ar sputter gun was used for the etching with the ion energy set at 200 eV and the middle range current was selected. The sputtering rate was estimated to be $\sim 0.01 \text{ nm s}^{-1}$.

As advised by the reviewer, we have re-examined our XPS data and provided Li₂O and LiF analysis for all studied electrolytes. Under the same magnification, the FST-SEI showed a large amount of Li₂O in the O1s spectra (**Figure R10, Figure 3b** of the revised manuscript), ascribing to the reduction of sulfolane reduction. However, a sharp decrease of Li₂O signal was observed in the FFT electrolytes and negligible Li₂O can be detected in the EE electrolytes. Instead, the Li₂CO₃ and LiOR signals increased largely for both FFT and EE electrolytes. This result validates that the FST electrolytes could promote the formation of Li₂O in the SEI by sulfolane reduction as suggested by the MD simulation (**Figure R11, Supplementary Figure 1** of the revised manuscript). A similar trend was found for the LiF signal among FST, FFT and EE electrolytes in the F1s spectra (**Figure R12, Supplementary Figure 11** of the revised manuscript). The simultaneous formation of Li₂O and LiF in FST electrolytes leads to the desired Li₂O-LiF composite SEI that will be beneficial for the long cycle of SiMPs.

Figure R10. The O1s spectra by XPS measurement on μ Si electrodes after 50 cycles in μ Si||Li cells with different electrolytes. All the XPS results were fitted with CasaXPS software. The binding energy was calibrated with C1s at 284.8 eV.

Figure R11. Reduction potentials from QC calculations using G4MP2 composite methodology and wB97XD/6-31+G(d,p) DFT calculations with all solvates immersed in implicit solvent modeled using PCM (ether) or SMD ($\epsilon=20$).

Figure R12. The F1s spectra by XPS measurement on μSi electrodes after 50 cycles in $\mu\text{Si}||\text{Li}$ cells with different electrolytes. All the XPS results were fitted with CasaXPS software. The binding energy was calibrated with C1s at 284.8 eV.

These results and discussions have been added to the revised manuscript (marked yellow below).

The revised XPS data collection part on page 29 of the revised manuscript:

The sample was directly moved from the Ar atmosphere to the XPS chamber with a vacuum transfer container to avoid exposure to the air. The neutralizer was applied during the data collection, and an Ar sputter gun was used for the etching with the ion energy set at 200 eV and the middle range current selected. The sputtering rate was estimated to be $\sim 0.01 \text{ nm s}^{-1}$. The etching procedure was carried out in a cycle of accumulated 0, 60, 120, 180, 300, and 600 seconds. Spectra were recorded of the sample surface before sputtering and between sputtering cycles.

The revised part of XPS analysis on pages 12–13:

Fig. 3 shows the SEI composition on the SiMP electrode after 50 plating/stripping cycles at 1 mA cm^{-2} and 4.1 mAh cm^{-2} in FST, FFT, and EE electrolytes (full spectra are shown in **Supplementary Figs. 11–13**). The outer and inner layer of SEI formed in EE and FFT electrolytes mainly consist of organic species (C-O/C=O peak, $\sim 286.5 \text{ eV}$, C-H/C-C peak, $\sim 284.8 \text{ eV}$) (**Fig. 3a**, **Supplementary Fig. 14a–b**). In comparison, the FST-SEI has a thinner C-H/C-C peak with a much weaker C-O/C=O intensity than that in EE/FFT electrolytes. Organic species were primarily found in the outer FST-SEI layer and disappeared after 300s sputtering while the inner layer of FST-SEI was almost exclusively $\text{Li}_2\text{O-LiF}$ (**Fig. 3b**, **Supplementary Fig. 14c**). In the O1s spectra, the FST-SEI showed a much higher Li_2O intensity compared to that in FFT-SEI, and only a negligible Li_2O signal was noticed in the EE-SEI (**Fig. 3b**). Instead, the Li_2CO_3 and LiOR signals increased largely for both FFT and EE electrolytes. This result validates that FST electrolytes could promote the formation of Li_2O in the SEI by sulfolane reduction as suggested by the MD simulation (**Supplementary Fig. 1**). A similar decrease trend was found for the LiF signal in the F1s spectra from FST to FFT and EE electrolytes (**Supplementary Fig. 11**). The simultaneous formation of Li_2O and LiF in FST electrolytes leads to the desired $\text{Li}_2\text{O-LiF}$ composite SEI that will be beneficial for the long cycle of SiMPs.

2. Figure 7 depicts the electrochemical performance of the $5\mu\text{m}$ silicon electrode in $\text{Li}/\mu\text{Si}$ half cells. However, the voltage profiles in Figure 7a-c are not well displayed. The reviewer suggests removing the areal capacity but highlighting the specific capacity of the μSi electrode in these

figures. In addition, the Li/ μ Si half cells are cycled between 0.05-1V, why the cut-off voltage is not selected from 0.01-1V?

Response: We thank the reviewer for the comment and nice suggestions.

As advised by the reviewer, we have removed the areal capacity in **Figure 7a–c** of the original manuscript and re-draw all figures with the highlight of μ Si specific capacity.

The Li|| μ Si half cells were cycled between 0.05 V and 1V for the following reasons: The capacity of the Si electrode at a potential between 0.05 V and 0.01 V is small due to the slopy charge/discharge curve. As shown in **Figure R13**, during the first lithiation process, we select the cut-off potential of 0.005 V to achieve full lithiation due to the low and flat lithiation potential. In the following cycles, we selected a 0.05 V cut-off potential to reflect the lithiation state in a full cell with an N/P ratio of 1.1.

Figure R13. The discharge curve of the Li|| μ Si (4.1 mAh cm⁻²) cells. The black line represents the first discharge curve with a cutoff potential of 0.005 V at C/20. The inset shows the corresponding released capacities at a cutoff voltage of 0.005, 0.01, and 0.05 V, respectively. The blue line indicates the following cycle at an improved rate of 0.25C with a cutoff voltage of 0.05V.

3. The authors showed that the μSi cycled in FST electrolyte “have a “crack-less” morphology and Si particles larger than 5 μm could still be noticed after 50 cycles” and no micro-sized particles were observed in the FIB cross-section of the electrodes for the reference electrolytes. Theoretically, μSi particle cracks will be inevitable during lithiation/de-lithiation, why the micro-sized morphology is retained in FST electrolytes? Moreover, why does μSi electrode swell largely in the first few cycles and is restrained after 20 cycles for FST electrolytes, but not the reference electrolytes?

Response: We thank the reviewer for the comments.

We agree with the reviewer that the μSi particles will crack during lithiation/de-lithiation, which will further fracture with continuous cycling if the SEI is not robust enough to prevent electrolyte penetration, as is commonly observed in traditional carbonate electrolytes. Our FST electrolytes could form a Si-phobic $\text{LiF}/\text{Li}_2\text{O}$ SEI as demonstrated with the XPS and ELLS data. This robust SEI could effectively inhibit electrolyte penetration and Si particle further fracture, which shows a “micro-sized” morphology after 200 lithiation/de-lithiation cycles in the $\text{Li}||\mu\text{Si}$ half cells even if the crystalline Si particle cracks to the amorphous phase (**Figure R14a**), compared to the porous structures with large pulverization found in the reference electrolytes (**Figure R14b–c**).

The μSi electrodes swelled largely in the first few cycles for all electrolytes due to the volume expansion of silicon particles during lithiation (**Figure R14d**). The Li_2O - LiF composite SEI formed from FST electrolytes is highly Si-phobic, which could maintain its integrity during silicon particle expansion (lithiation) and contraction (de-lithiation), effectively restraining Si particle swelling in the following cycles. In addition, the integrity of Li_2O - LiF SEI could also prevent electrolyte penetration, therefore, the thin SEI layer formed through the first few cycles will be robust enough to allow the inner μSi electrode reversible expansion/contraction. All the above factors lead to a restrained electrode swelling in FST electrolytes after the first 20 cycles (**Figure R14d**, magenta). On the other hand, for the carbonate EE electrolytes, the Si-philic organic-rich SEI adheres to the Si particle strongly and suffers the same mechanical tension during the lithiation/delithiation cycles, which cracks easily with SiMPs. This will allow the electrolytes to penetrate the cracked SiMPs forming SEI and further separating the pulverized Si. The formed SEI in cracked SiMPs further increased the volume change of lithiated Si. The continuous Si pulverization and SEI formation result in continuous swelling of the μSi electrode (**Figure R14d**,

black). The fluorinated FFT electrolytes could form additional Si-phobic LiF in the SEI through fluorinated solvent reduction, which leads slower electrode swelling rate compared to EE electrolytes. However, the SEI formed from FFT electrolytes is still organic abundant, largely limiting the robustness of the SEI, therefore, a large electrode swelling is still observed along the cycling (Figure R14d, blue).

Figure R14. Morphology of Si particles and electrode thickness after 200 cycles. a–c, Focused ion beam (FIB) cross-section SEM images of the SiMP electrodes after 200 cycles of operation in different electrolytes. a, FST. b, FFT. c, EE. d, The electrode thickness evolution during the cycling with various electrolytes.

Figures R14 were added to Figure 7, and these results and discussions have been updated in the revised manuscript (marked in yellow below).

The revised part on pages 20–21 of the revised manuscript:

As shown in Fig. 7a–c, the SiMPs after charge/discharge in FST electrolytes for 200 cycles showed “crack-less” morphology (Fig. 7a), similar to the crack-free pristine Si with expended size and

deformed shape (**Supplementary Fig. 23b**). Only minor fractures were found in the SiMPs electrode. In addition, a homogeneous distribution of C, O, and F was identified in the elemental energy-dispersive X-ray spectroscopy (EDX) mapping (**Supplementary Fig. 27**), validating a uniform Li₂O-LiF SEI layer formation. In sharp contrast, large fractures with huge porous structures have developed in SiMPs cycled with the reference electrolytes (**Fig. 7b** for FFT, **7c** for EE). The C, O, and F elements were found unevenly spreading over the electrode with a high intensity of C and O for the SiMPs cycled with FFT and EE electrolytes (**Supplementary Figs. 28–29**), correlating to the organic-rich SEI that leads to continuous electrolyte penetration and further SiMP pulverization. The large pores in the swelled μ Si electrode lead to the loss of contact between active SiMPs and carbon black, resulting the fast capacity decay.

4. Minor issues: 1) Figure labelings: In Figure 6, the labelings should be unified with the same font and text size; in Figures 8 and 9, the resolution labelings are too small to be recognized, they should be enlarged with better resolution; 2) The electrolyte flammability test should be provided with recorded videos, a single image can not tell the full examination of the investigation; 3) In the full-cell data of Fig.10d, why does the capacity fluctuate during cycles? Lastly, 12 figures are exceptional for an article, the authors are suggested to combine the figures or move some figures to the supporting information.

Response: We thank the reviewer for his/her careful observation of our data. These issues have been addressed carefully as follows:

1) The text font and size have been unified and enlarged for all Figures, including Figures 6, 8, and 9 in the original manuscript.

2) We have provided full recorded videos for the electrolytes' flammability tests, and the description was added to **Supplementary Figure 6 (Figure R15 below)** and **Supplementary Note 1** on page 4 of the revised manuscript (marked in yellow below).

Figure R15. Flammability test for the three different electrolytes: EE (a), FFT (b), and FST (c). These images are extracted from the supplementary videos 1 (EE), 2 (FFT), and 3 (FST).

The revised part on page 10 of the revised manuscript:

Moreover, because of the flame retardant nature of SL molecule,³⁹ the FST electrolytes are not flammable and offer improved safety benefits as the FFT electrolytes (Supplementary Note 1, Supplementary Fig. 6, Supplementary videos 1–3).^{29,40}

The revised part on page 4 of the revised Supplementary Note 1:

As demonstrated in flaming tests (Supplementary Fig. 6, Supplementary videos 1–3 for EE, FFT, and FST electrolytes, respectively), this FST electrolyte does not burn following ignition, in sharp contrast with the highly flammable EE electrolyte. This is attributed to two facts: one is the fluorination on FEC, TTE molecules, which can effectively serve as an inhibitor to the propagation of oxygen radicals during combustion;⁹⁻¹⁰ second is the SL itself can act as flame retardant.^{7,8}

3) The jumbled capacity in Fig. 10d of the original manuscript is probably a result of temperature fluctuations, which is commonly observed in uncontrolled temperature environments (*Nature Energy*, 2022, 7, 94-106).

Lastly, we thank the reviewer for his/her nice suggestion and have moved **Figures 2**, **Figure 5c**, and **Figure 10b–c** to the Supporting Information, and merged **Figures 3 and 4**, **Figures 10 and 11**, with nine primary figures shown in the revised manuscript. We can make further layout cutoffs based on the requirement of article printing once accepted in the journal.

REVIEWERS' COMMENTS

Reviewer #1 (Remarks to the Author):

The authors faithfully revised the manuscript according to the reviewer's suggestions and comments. At this stage, it contains broad interest for the reader to understand all the contents. Without further modifications, it can be accepted.

Reviewer #2 (Remarks to the Author):

The authors have addressed the comments and the paper is now suitable for publication.